# Unifying Deep Stochastic Processes for Image Enhancement

**Wojciech Kozłowski** [1]   **Radosław Kuczbański** [1]   **Kamil Adamczewski** [1]   **Karol Szczypkowski** [1]   **Maciej Zieba** [1 2]

## Abstract

Deep stochastic processes have recently become a central paradigm for image enhancement, with many methods explicitly conditioning the stochastic trajectory on the degraded input. However, the relationship between these conditional processes and standard diffusion models remains unclear. In this work, we introduce a unified perspective on stochastic image enhancement by classifying recent methods into three families of continuous-time processes: unconditional diffusion models, Ornstein–Uhlenbeck (OU) processes, and diffusion bridges. We show that all of these approaches arise from a common stochastic differential equation (SDE) formulation. This framework makes explicit that seemingly disparate methods differ primarily in their drift and diffusion terms, terminal distributions, and boundary conditions, while schedulers and samplers constitute orthogonal design choices. Leveraging this unification, we conduct a controlled empirical study across multiple image enhancement tasks using identical architectures and training protocols. Our results reveal no consistently dominant method; instead, we identify and disentangle the specific design choices that most strongly influence performance. Finally, we release ItoVision, a modular PyTorch library that implements the unified framework and enables rapid prototyping and fair comparison of stochastic image enhancement methods.

## 1. Introduction

Image enhancement encompasses a broad class of inverse problems—such as super-resolution, low-light enhancement, colorization, and deraining—in which the goal is to recover a high-quality image from a degraded observation. These problems are inherently ill-posed: a single low-quality input

typically corresponds to a distribution of plausible high-quality reconstructions. As a result, modern image enhancement systems must go beyond point estimation and instead model complex conditional distributions, making generative modeling a natural and increasingly dominant paradigm (Ledig et al., 2016; Saharia et al., 2022a;b).

Diffusion-based models have recently emerged as a strong baseline for image enhancement, offering stable training and high perceptual quality (Ho et al., 2020; Saharia et al., 2022a;b; Rombach et al., 2022). Motivated by the structure of restoration problems, a growing body of work has proposed alternative stochastic processes that explicitly condition the generative trajectory on the degraded input. These methods (Yue et al., 2023b; Delbracio & Milanfar, 2023; Li et al., 2023; Liu et al., 2023; Luo et al., 2023a; Zhou et al., 2023b; Yue et al., 2023a; Zhu et al., 2025) aim to interpolate between low- and high-quality images, rather than evolving from unconditional noise, as illustrated in Figure 1. Intuitively, such conditional trajectories promise to reduce uncertainty, restrict the modeled space, and focus learning on recovering missing details.

However, despite their conceptual appeal, the actual benefits of these conditional stochastic processes remain poorly understood. Existing methods differ not only in their stochastic definitions, but also in schedulers, samplers, discretization strategies, and backbone architectures. As a consequence, reported improvements are often entangled with implementation choices, making it difficult to isolate the effect of the underlying process itself. In practice, these methods are rarely compared directly, as they are usually considered to come from different frameworks, despite their underlying mathematical structure being very similar (Yue et al., 2023b; Zhou et al., 2023b; Zhu et al., 2025). A key obstacle to fair comparison is that method definitions are typically inseparable from their original schedulers and samplers. In this work, we show that this coupling is largely artificial.

In this work, we introduce a unified perspective on stochastic image enhancement by explicitly classifying recent methods into three families of continuous-time processes: unconditional diffusion models, Ornstein–Uhlenbeck (OU) processes, and diffusion bridges. By reformulating these methods in continuous time, we show that nearly all modern stochastic image enhancement approaches can be ex-

---

[1]Wrocław University of Science and Technology , Wrocław, Poland [2]Tooploox, Warsaw, Poland. Correspondence to: Wojciech Kozłowski <wojciech.kozlowski@pwr.edu.pl>.

*Proceedings of the $43^{rd}$ International Conference on Machine Learning*, Seoul, South Korea. PMLR 306, 2026. Copyright 2026 by the author(s).

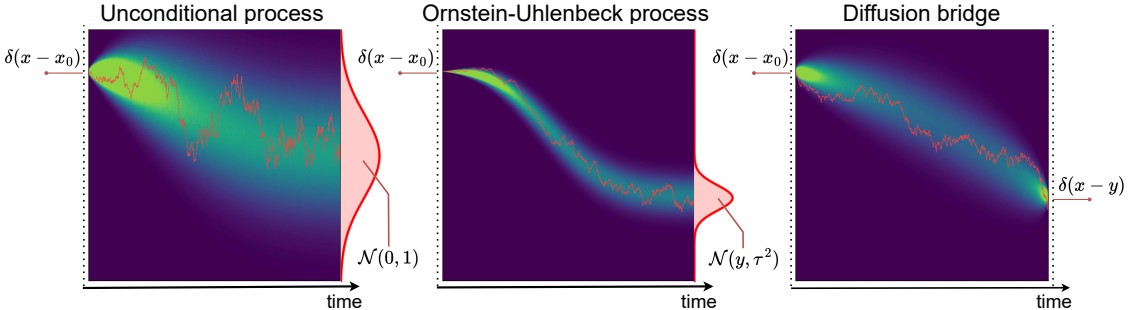

*Figure 1.* 1D visualization of three classes of considered methods and how conditioning with y affects them. Left: Unconditional processes gradually perturb $x_0$ into Gaussian noise $\mathcal{N}(0, 1)$, independently of $y$. Middle: Ornstein–Uhlenbeck processes converge to a terminal distribution centered at $y$ with variance $\tau^2$. Right: Diffusion bridges start at $x_0$ and are conditioned to reach $y$ at the terminal time. For ease of visualization, we consider fixed starting point for each process.

pressed as instances of a single stochastic differential equation (SDE), differing only in their drift and diffusion terms, terminal distributions, and boundary conditions. Importantly, schedulers, samplers, and discretization schemes emerge as orthogonal design choices rather than defining characteristics of the method.

The proposed framework generalizes several existing methods and clarifies their theoretical relationships. In particular, discrete Markov chain–based image enhancement methods (Yue et al., 2023b) can be interpreted as marginalizations of continuous-time stochastic processes, allowing evaluation at arbitrary intermediate times without altering their original behavior. Likewise, flow-based models (Lipman et al., 2022; Liu et al., 2022) admit equivalent stochastic differential equation formulations whose reverse-time ordinary differential equations recover the original flows in distribution and, under appropriate solvers, even at the trajectory level. Finally, we show that Brownian and Schrödinger bridge–based methods (Li et al., 2023; Liu et al., 2023; Zhou et al., 2023b) arise naturally through Doob's $h$-transform (Doob, 1984), revealing that several approaches previously treated as distinct are, in fact, special cases or reparameterizations of a common underlying process. This perspective exposes previously overlooked equivalences and helps disentangle genuine modeling differences from implementation choices.

This unified formulation allows us to conduct a large-scale, fully controlled empirical study of stochastic image enhancement methods. Using identical backbones, losses, and training protocols across multiple tasks, we find that most methods achieve remarkably similar performance. In many cases, plain diffusion models match or even outperform more specialized OU- or bridge-based approaches, consistent with observations in other generative modeling domains where architectural and training choices dominate method-specific differences (Lucic et al., 2018; Karras et al., 2022). These results challenge the common assumption that conditioning

the stochastic trajectory inherently leads to better restoration quality.

Crucially, we go beyond surface-level benchmarking and analyze why certain methods underperform. We identify low-temperature settings in OU-based methods as a key failure factor, showing that they induce strong collinearity between intermediate states and the input, which limits detail recovery (Delbracio & Milanfar, 2023; Luo et al., 2023c). We further demonstrate that deterministic samplers, while popular for diffusion bridges (Yue et al., 2023a; Zhou et al., 2023b), systematically cause oversmoothing by eliminating the only source of stochasticity in these models.

Finally, we release a unified and extensible PyTorch library that directly implements the proposed framework, enabling reproducible comparisons and modular experimentation, in the spirit of recent reproducibility-focused efforts in diffusion modeling (von Platen et al., 2022). By standardizing both theory and practice, this work establishes a new baseline for fair and interpretable evaluation of stochastic image enhancement methods.

Our contributions are fourfold: (1) We propose a unified continuous-time SDE framework that generalizes unconditional models, including flow-based methods, Ornstein–Uhlenbeck processes, and diffusion bridges, decoupling method definitions from schedulers, samplers, and discretization strategies. (2) Using this framework, we conduct a large-scale, controlled empirical study across multiple image enhancement tasks with identical architectures and training protocols, establishing a new standard for fair comparison. (3) We identify concrete performance-limiting factors—such as low-temperature–induced collinearity and oversmoothing caused by deterministic sampling—and provide actionable insights via extensive ablations on samplers and discretization schemes. (4) We release a unified, modular PyTorch library that implements the proposed framework, and supports future method development.

## 2. Unified framework

In this section, we formulate a unified mathematical framework for stochastic image enhancement by expressing a broad range of existing methods as continuous-time stochastic processes. We first introduce the general stochastic differential equation (SDE) formulation that underlies all considered approaches. We then analyze representative methods from the literature, which we grouped into three families—unconditional processes, Ornstein–Uhlenbeck processes, and diffusion bridges—and show how each can be recovered as a special case of the unified framework. Finally, we present a modular implementation that instantiates all unified formulations within a single library. Throughout this section, we use propositions and remarks to highlight key contributions of the paper.

### 2.1. Unified Method Definitions

A central contribution of this work is the unified formulation of modern stochastic image enhancement methods summarized in Table 1. This work provides a standardized representation of each method by explicitly separating the definition of the forward stochastic process from implementation-dependent choices such as schedulers, samplers, and discretization strategies.

Each method is described through three components: (i) the forward process, (ii) the transition kernels induced by this process, and (iii) the base distribution used to initialize reverse-time generation.

**Forward process** denoted as $(\mathbf{x}_t \in \mathbb{R}^d)_{t \in [0,1]}$ is fully defined by the forward stochastic differential equation (SDE), the initial distribution, and the distribution of the conditioning variable:

$$d\mathbf{x}_t = \mathbf{f}(\mathbf{x}_t, t, \mathbf{y}) \, dt + g(t) \, d\mathbf{w}_t, \qquad (1)$$
$$\mathbf{x}_0 \sim p_{data}(\mathbf{x}_0), \quad \mathbf{y} \sim p_{low}(\mathbf{y}|\mathbf{x}_0),$$

where the SDE model is defined by a deterministic drift $\mathbf{f} : \mathbb{R}^d \times [0,1] \times \mathbb{R}^d \to \mathbb{R}^d$ , and a diffusion coefficient $g : [0,1] \to \mathbb{R}$, $\mathbf{w}_t \in \mathbb{R}^d$ is a $d$-dimensional Wiener process, $\mathbf{x}_0 \in \mathbb{R}^d$ is $d$-dimensional random variable denoting high quality data, and $\mathbf{y} \in \mathbb{R}^d$ denotes corresponding low-quality images. The model needs to be constructed so that at the terminal time $t = 1$ the process converges to its base distribution $p_1$ which should be easy to sample from. Note that $\mathbf{y}$ depends only on $\mathbf{x}_0$ and $\mathbf{x}_0$ is independent of Brownian motion, therefore, we can still consider (1) in the Ito sense.

**Transition kernel** is the conditional distribution induced by the forward SDE at time $t$ given $\mathbf{x}_0$ and $\mathbf{y}$. The transition kernel is derived as:

$$p(\mathbf{x}_t|\mathbf{x}_0, \mathbf{y}) = \mathcal{N}(\mathbf{x}_t; \boldsymbol{\mu}_t(\mathbf{x}_0, \mathbf{y}), \sigma_t^2 \mathbb{I}), \qquad (2)$$

with specific functions $\boldsymbol{\mu}_t$ and $\sigma_t^2$ for each method.

**Base distribution** that can be treated as a transition kernel at the terminal time $t = 1$. The base distribution is indicated as $p_1(\mathbf{x}_1|\mathbf{y})$ and it can be the Gaussian or Dirac delta depending on the method.

In order to generate a new high-quality image $\hat{\mathbf{x}}$, we need to first sample $\mathbf{x}_1 \sim p_1(\mathbf{x}_1|\mathbf{y})$ and reverse (1). Thanks to (Anderson, 1982; Zhang & Chen, 2021)[1] we have another SDE, where time goes backward and it almost surely goes through $\mathbf{x}_0$ at time $t = 0$:

$$d\mathbf{x}_t = [\mathbf{f}(\mathbf{x}_t, t, \mathbf{y}) - \frac{\lambda^2 + 1}{2} g(t)^2 \nabla_{\mathbf{x}_t} \log p_t(\mathbf{x}_t|\mathbf{y})] \, dt +$$
$$\lambda g(t) \, d\bar{\mathbf{w}}_t, \qquad (3)$$
$$\mathbf{x}_1 \sim p_1(\mathbf{x}_1|\mathbf{y}), \quad \mathbf{y} \sim p_{low}(\mathbf{y}),$$

where $\bar{\mathbf{w}}_t \in \mathbb{R}^d$ is a d-dimensional reversed Wiener process and $\lambda \in \mathbb{R}^+ \cup \{0\}$ is a nonnegative parameter that controls the level of stochasticity of the reverse process. In principle, (3) can be solved numerically using standard samplers such as Euler–Maruyama. In practice, however, this requires access to $\log p_t(\mathbf{x}_t|\mathbf{y})$, which is intractable as it involves integration over $p_{data}$. Instead, we approximate its gradient with a neural network $v_\theta$ with learnable parameters $\theta$ such that:

$$\theta^* = \arg \min_{\theta} \mathbb{E} || v_\theta(\mathbf{x}_t, t, \mathbf{y}) - \nabla_{\mathbf{x}_t} \log p_t(\mathbf{x}_t|\mathbf{x}_0, \mathbf{y}) ||^2,$$

which is known as the score matching loss function (Hyvärinen & Dayan, 2005; Vincent, 2011; Song & Ermon, 2019).

For clarity, we organize the methods into three broad families: standard diffusion models, Ornstein–Uhlenbeck (OU) processes, and diffusion bridges. Importantly, all methods are defined independently of the specific noise scheduler $(\beta_t > 0)_{t \in [0,1]}$. We denote the Riemann integral of the scheduler by $\alpha_{s,t} = \int_s^t \beta_z \, dz$ and define $\phi_{s,t} = \exp(-\alpha_{s,t})$. For notational convenience, we use $\alpha_t \equiv \alpha_{0,t}$ and $\phi_t \equiv \phi_{0,t}$. The temperature parameter is denoted by $\tau$ across all methods.

We define method equivalence in terms of a shared SDE formulation of the forward process, and consider schedulers, discretizations, samplers, and similar components to be method-independent choices. This decoupling of stochastic process definitions from numerical choices enables direct and fair comparison between methods that were previously treated as separate frameworks. The specific implementation settings used in our experiments, including schedulers, samplers, and temperature values, are summarized in Table 3.

---

[1]For complete proof, see (Zhang & Chen, 2021) Appendix H.

## 2.2. Unconditional processes

This class of models were originally designed as fully generative models; as a result, the input observation $\mathbf{y}$ does not appear in the definitions of the forward processes, transition kernels, or base distributions. To obtain a conditional model capable of generating an enhanced version of $\mathbf{y}$, the observation must instead be provided as an additional input to the backbone network $v_\theta$ (Rombach et al., 2022; Saharia et al., 2022b; Jinhui et al., 2023). A prominent example of this class is the Diffusion Model (DM) (Sohl-Dickstein et al., 2015; Ho et al., 2020), which was initially formulated as a discrete Markov process that progressively transforms the data distribution into Gaussian noise. This framework was later reformulated in continuous time by (Song et al., 2020b) into two variants—Variance Exploding (VE) and Variance Preserving (VP)—summarized in Table 1. In the VE formulation, the expected value of the transition kernel remains constant while its variance grows unbounded, whereas in the VP formulation the drift term $\mathbf{f}(\mathbf{x}_t, t) = -\beta_t \mathbf{x}_t$ gradually pushes $\mathbf{x}_t$ toward zero with a strength controlled by the scheduler $\beta_t$. In a similar way as diffusion model, below we show that flow matching (Lipman et al., 2022; Liu et al., 2022; Geng et al., 2026) can also be presented in the form of SDE.

**Flow Matching** is derived from normalizing flow theory and formulates generative modeling as the regression of a deterministic vector field that transports one probability distribution to another, guided by optimal transport.

In this work, we show that Flow Matching can also be expressed as the continuous stochastic process for which the reverse ODE (Eq. (3) with $\lambda = 0$) recovers the original procedure from (Lipman et al., 2022; Liu et al., 2022; Geng et al., 2026).

**Proposition 2.1.** *Let $\mathbf{x}_t$ be a continuous stochastic process that follows the SDE*

$$\mathrm{d}\mathbf{x}_t = -\beta_t \mathbf{x}_t \, \mathrm{d}t + \sqrt{2(1-\phi_t)\beta_t} \, \mathrm{d}\mathbf{w}_t. \tag{4}$$

*If the scheduler is defined as $\beta_t = \frac{1}{1-t}$, then for any $\mathbf{x}_0 \sim p_{data}(\mathbf{x}_0)$ and $\epsilon \sim \mathcal{N}(\mathbf{0}, \mathbb{I})$ we have $\mathbf{x}_t = (1-t)\mathbf{x}_0 + t\epsilon$ for $t \in [0, 1)$, which corresponds to transitions of (Liu et al., 2022).*

*Proof sketch.* Eq. (4) admits the mild solution:

$$\mathbf{x}_t = \phi_t \mathbf{x}_0 + \int_0^t \phi_{s,t} g(s) \mathrm{d}\mathbf{w}_s. \tag{5}$$

The deterministic term specifies the conditional mean $\mathbb{E}[\mathbf{x}_t|\mathbf{x}_0] = \phi_t \mathbf{x}_0$ while the stochastic integral accounts for the variance. By Ito isometry,

$$\mathrm{Var}(\mathbf{x}_t) = \int_0^t \phi_{s,t}^2 g^2(s) ds, \tag{6}$$

which can be computed in closed form and yields $\mathrm{Var}(\mathbf{x}_t) = (1 - \phi_t)^2$. For $\beta_t = \frac{1}{1-t}$, we have $\phi_t = 1 - t$, and therefore:

$$\mathbf{x}_t = (1-t)\mathbf{x}_0 + t\epsilon, \quad \epsilon \sim \mathcal{N}(\mathbf{0}, \mathbb{I}). \tag{7}$$

With Proposition 2.1, we can reverse Eq. (4) using the Euler method to recover the original implementation of (Liu et al., 2022)[2]. At the same time, this formulation allows the reverse-time process to be sampled with different standard samplers. We include this method in the comparison to measure its effectiveness against conditioned processes.

## 2.3. Ornstein-Uhlenbeck processes

The second class of processes considered in our framework consists of generalized Ornstein–Uhlenbeck (OU) processes, defined by the stochastic differential equation:

$$\mathrm{d}\mathbf{x}_t = \beta_t(\boldsymbol{\mu} - \mathbf{x}_t)\mathrm{d}t + g(t)\mathrm{d}\mathbf{w}, \tag{8}$$

where $\beta_t > 0$ and $g(t) \geq 0$ are time-dependent functions. This process converges to a Gaussian stationary distribution with mean $\boldsymbol{\mu}$ and variance governed by $g(t)$. Notably, both diffusion and flow-matching models can be interpreted as special cases of OU processes with $\boldsymbol{\mu} = \mathbf{0}$.

In this work, we focus on OU processes whose stationary distribution is $p_1(\mathbf{x}_1 \mid \mathbf{y}) = \mathcal{N}(\mathbf{x}_1; \mathbf{y}, \tau^2\mathbf{I})$, and accordingly identify the conditioning image $\mathbf{y}$ with the mean parameter, i.e., $\boldsymbol{\mu} \equiv \mathbf{y}$. A representative example is IR-SDE (Luo et al., 2023a), which corresponds to a mean-reverting OU process with a diffusion coefficient similar to that used in standard diffusion models. In the original work, the method uses a discrete scheduler, that is the integral of the function $\beta(t)$ was calculated numerically. This approach limits both speed and flexibility, as the values of $\int_0^{t_i} \beta(s)ds$ must be recomputed for each discrete timestep $\{t_i\}_{i=0}^{K}$. In contrast, we derive a closed-form analytical solution that supports arbitrary step counts without parameter recomputation (see Appendix B). Below, we derive the formulations of ResShift (Yue et al., 2023b) and InDI (Delbracio & Milanfar, 2023) within the framework of continuous stochastic processes, showing that both can be interpreted as Ornstein–Uhlenbeck processes.

**ResShift** was originally proposed as a discrete-time model defined by a Markov chain that interpolates between a high-quality image $\mathbf{x}_0$ and a Gaussian distribution centered at the degraded image $\mathbf{y}$ with variance $\tau^2$. The one-step forward transition is given by:

$$p(\mathbf{x}_t|\mathbf{x}_{t-1}, \mathbf{y}) = \mathcal{N}(\mathbf{x}_t; \mathbf{x}_{t-1} + \delta_t \mathbf{e}, \tau^2 \delta_t \mathbb{I}), \tag{9}$$

---

[2]We do not consider rectification procedure.

where $\mathbf{e} = \mathbf{y} - \mathbf{x}_0$ denotes the residual between the low- and high-quality images, $\delta_t$ is a discrete noise scheduler, and $t \in \{0, \ldots, T\}$ indexes the diffusion steps. Marginalizing over intermediate steps yields a closed-form transition from $\mathbf{x}_0$ to an arbitrary timestep $t$:

$$p(\mathbf{x}_t|\mathbf{x}_0, \mathbf{y}) = \mathcal{N}(\mathbf{x}_t; \mathbf{x}_0 + \eta_t\mathbf{e}, \tau^2\eta_t\mathbb{I}), \quad (10)$$

where $\eta_t = \sum_{i=0}^{t} \delta_i$.

**Proposition 2.2.** *The Ornstein–Uhlenbeck process defined by the stochastic differential equation*

$$\mathrm{d}\mathbf{x}_t = \beta_t(\mathbf{y} - \mathbf{x}_t)\,\mathrm{d}t + \tau\sqrt{\beta_t(2 - \phi_t)}\,\mathrm{d}\mathbf{w}_t \quad (11)$$

*induces the same transition densities as the original ResShift model in Eq. (10) for $t \in [0, 1]$.*

*Proof sketch.* To generalize ResShift to continuous time, we define $\phi_t = 1 - \eta_t$ and rewrite the residual explicitly as $\mathbf{e} = \mathbf{y} - \mathbf{x}_0$, turning Eq. (10) into:

$$p(\mathbf{x}_t|\mathbf{x}_0, \mathbf{y}) = \mathcal{N}(\mathbf{x}_t; \phi_t\mathbf{x}_0 + (1-\phi_t)\mathbf{y}, \tau^2(1-\phi_t)\mathbb{I}). \quad (12)$$

We then seek a continuous-time stochastic process whose transition kernels match Eq. (12). Eq. (11) admits a mild solution of the form

$$\mathbf{x}_t = \phi_t\mathbf{x}_0 + \mathbf{y}\int_0^t \phi_{s,t}\beta_s\,\mathrm{d}s + \int_0^t \phi_{s,t}g(s)\,\mathrm{d}\mathbf{w}_s, \quad (13)$$

where $\mathbf{y}$ is treated as an independent conditioning variable. The deterministic part evaluates to $\mathbb{E}[\mathbf{x}_t] = \phi_t\mathbf{x}_0 + (1 - \phi_t)\mathbf{y}$, and $\mathrm{Var}(\mathbf{x}_t) = \int_0^t \phi_{s,t}^2 g(s)^2\mathrm{d}s$. By choosing $g(t)^2 = \tau^2\beta_t(2-\phi_t)$ the integral evaluates to $\tau^2(1-\phi_t)$, recovering the transition kernel in Eq. (12). A complete derivation is provided in Appendix A.2.

Proposition 2.2 clarifies how ResShift relates to other OU-based methods, such as IR-SDE (Luo et al., 2023c), by showing that their differences arise from the specific choice of diffusion coefficients, while both processes share the same drift and stationary distribution. This continuous-time reformulation also enables fair experimental comparisons by decoupling the method definition from discretization choices.

**InDI** defined a continuous process with the following formula for intermediate samples:

$$\mathbf{x}_t = (1 - t)\mathbf{x}_0 + t\mathbf{y} + \tau t\epsilon, \quad \epsilon \sim \mathcal{N}(\mathbf{0}, \mathbb{I}). \quad (14)$$

The generative process starts with $\mathbf{x}_1 \sim \mathcal{N}(\mathbf{x}_1; \mathbf{y}, \tau^2\mathbb{I})$ and solves the deterministic ODE using the Euler method.

In this work, we show that this method can be formulated as an OU process. It can be viewed as a generalization of Flow Matching (Lipman et al., 2022; Liu et al., 2022) to arbitrary Gaussian priors, in the same spirit as IR-SDE (Luo et al., 2023c) generalizes diffusion models (Ho et al., 2020).

**Proposition 2.3.** *The OU-process with the following stochastic differential equation:*

$$\mathrm{d}\mathbf{x}_t = \beta_t(\mathbf{y} - \mathbf{x}_t)\,\mathrm{d}t + \tau\sqrt{2\beta_t(1 - \phi_t)}\,\mathrm{d}\mathbf{w}_t. \quad (15)$$

*If the scheduler is defined as $\beta_t = \frac{1}{1-t}$, then for any $\mathbf{x}_0 \sim p_{data}(\mathbf{x}_0)$ and $\epsilon \sim \mathcal{N}(\mathbf{0}, \mathbb{I})$ we have the same marginals as (14) for $t \in [0, 1)$.*

*Proof sketch.* Similarly to ResShift, the Eq. (15) yields the solution described in Eq. (13). However, due to the differences in the diffusion term $g(t)$ the analytical solution is defined as:

$$p(\mathbf{x}_t|\mathbf{x}_0, \mathbf{y}) = \mathcal{N}(\mathbf{x}_t; \phi_t\mathbf{x}_0 + (1 - \phi_t)\mathbf{y}, \tau^2(1 - \phi_t)^2\mathbb{I}). \quad (16)$$

The proof can be found in Appendix A.3. With proposition 2.3 we can use the reversed scheduler $\beta_t = \frac{1}{1-t}$ with the Euler sampler of reversed (3) with $\lambda = 0$ to recover the original implementation. Similarly to previous propositions, the choice of sampler is independent of the model definition, and therefore we evaluate several variants.

For both ResShift and InDI, the derived form of the forward processes, transition kernels, and base distributions are summarized in Table 1.

### 2.4. Diffusion bridges

The final class of models modifies the forward SDE to ensure that the process reaches $\mathbf{y}$ at the terminal time $t = 1$ almost surely. For any SDE with a form

$$\mathrm{d}\mathbf{x}_t = \mathbf{f}(\mathbf{x}_t, t)\,\mathrm{d}t + g(t)\,\mathrm{d}\mathbf{w}_t, \quad (17)$$

one can obtain this conditioning by applying the h-transform (Doob, 1984)

$$\mathrm{d}\mathbf{x}_t = [\mathbf{f}(\mathbf{x}_t, t) + g^2(t)\nabla_{\mathbf{x}_t}\log p_1(\mathbf{y}|\mathbf{x}_t)]\,\mathrm{d}t + g(t)\,\mathrm{d}\mathbf{w}_t. \quad (18)$$

Applying this construction to the VE and VP diffusion models with standard noise schedules results in the Denoising Diffusion Bridge Models (DDBM-VE and DDBM-VP) (Zhou et al., 2023b).

Similarly, GOUB (Yue et al., 2023a) applies the $h$-transform to an Ornstein–Uhlenbeck process with the same parameterization as IR-SDE; Finally, UniDB (Zhu et al., 2025) generalizes diffusion bridges through a stochastic optimal control perspective, introducing a terminal cost weighted by $\gamma$ that penalizes deviation from the target $\mathbf{y}$ while regularizing trajectory length; in the limit $\gamma \to \infty$, this formulation recovers the classical $h$-transform, whereas finite $\gamma$ allows a controlled trade-off between endpoint accuracy and path smoothness. Following (Zhu et al., 2025), we adopt the same formulation and choice of $\gamma$.

Similarly to IR-SDE, for GOUB and UniDB we derive a closed-form solution of the cosine scheduler, which was computed numerically in the original papers. This approach improves both speed and flexibility, allowing the scheduler to operate with arbitrary timesteps during inference.

Below, we discuss the BBDM (Li et al., 2023) and I²SB (Liu et al., 2023) methods, and through the perspective of the h-transform, we show that they have an equivalent definition to the DDBM-VE process.

**Brownian Bridge Diffusion Model (BBDM)** defines image-to-image translation as a stochastic Brownian bridge diffusion process that translates between the source and target domains.

**Remark 2.4.** The Brownian bridge definition can be seen as the result of applying the h-transform on the Wiener process. The Wiener process can be generalized to the diffusion VE variant with a constant scheduler $\beta_t = 1$. Therefore, DDBM-VE and this model have the same definition but different scheduler choice.

**I²SB** approaches the bridge problem from a different perspective. Their theory is rooted in the findings on dynamic Schrodinger bridges (SB) from (Chen et al., 2021), but they considered the scenario, where we can sample from the joint distribution $p_{data,low}(\mathbf{x}, \mathbf{y})$ during training, which is generally not required for SB models.

**Remark 2.5.** Deriving the bridge process from (Liu et al., 2023) Theory 3.1 with $\mathbf{f}(\mathbf{x}_t, t) = \mathbf{0}$ and $g(t) = \sqrt{\beta_t}$ leads to the same result as applying h-transform (Doob, 1984) to the exact same SDE $d\mathbf{x}_t = \sqrt{\beta_t}\, d\mathbf{w}_t$. This gives the definition of DDBM-VE (Zhou et al., 2023b).

We elaborate on Remark 2.5 in Appendix A.4. As a result, we show that BBDM, DDBM-VE, and I²SB, although derived from different theoretical frameworks and often regarded as fundamentally incomparable (Yue et al., 2023a), are in fact the same method, differing only in their choice of scheduler, as illustrated in Table 1. Please also refer to Appendix F for elaboration on methods not included in our study.

### 2.5. Library

Building on the unified mathematical framework that underlies all considered methods, we release *Ito Vision*, a Python library implemented in PyTorch. The library combines the tested approaches into a single modular framework. This ensures consistency with the theoretical formulation and enables faithful reproduction of our results. Beyond serving as a reference implementation, *Ito Vision* is designed to be easily extensible: by following the unified framework summarized in Table 1, researchers can prototype and integrate new methods with minimal effort. We hope this library will

provide a practical foundation for future research and accelerate the development of novel approaches in this domain. Further details are provided in Appendix C.

## 3. Experiments

In this section, we report experiments on four image enhancement tasks: super-resolution, low-light enhancement, colorization, and deraining. We first describe the training setup, then compare the methods, and explain the differences. Finally, we show how different samplers and discretization techniques influence the results.

### 3.1. Experimental setup

For different tasks, we use backbones of varying sizes to study the scalability of the methods, considering both UNet-based architectures and Diffusion Transformers (DiT) (Peebles & Xie, 2022). Within each task, however, all methods share the same backbone architecture, model size, and dataset setup to ensure a fair comparison. Unless otherwise specified, we evaluate methods using their default settings from the original works, while fixing the network parameterization across models and adopting $\mathbf{x}_0$ prediction, which we found to perform best across all methods. To isolate the effect of the underlying process definitions on final performance, all models are trained using mean squared error (MSE) as a common loss function. In addition to this unified evaluation setup, we also report results obtained using the original configurations of each method, including their native parameterizations, loss functions, and samplers.

We evaluate single-image super-resolution on two benchmarks: FFHQ (Karras et al., 2019) downsampled by a factor of 8 ($64 \times 64 \rightarrow 512 \times 512$) and DIV2K (Agustsson & Timofte, 2017). Low-light image enhancement is conducted on the LOL dataset (Wei et al., 2018), image colorization on ImageNet (Russakovsky et al., 2015) using a latent diffusion setup, and image deraining on Rain1400 (Fu et al., 2017). Detailed experimental settings for each task are provided in Appendix D and Table 4.

### 3.2. Methods comparison

We compare unconditional, Ornstein–Uhlenbeck (OU), and diffusion-bridge processes, with quantitative results shown in Table 2. More detailed results can be found in Appendix in Tables 5, 6, 7, 8, and 9. Qualitative comparisons of restored output images are in Figures 6, 7, 8, and 9 in Appendix. Both quantitative metrics and visual results indicate that, when trained under the same protocol, all methods achieve comparable performance. Further statistical analysis shows a slight advantage for ResShift followed by the diffusion model, as shown in the aggregated results in Table 10 in Appendix. Analyzing results by process family

*Table 1.* Different design choices of forward SDE and transition densities of iteration-based image enhancement methods. We organize the methods into three groups: unconditional processes, OU processes and diffusion bridges. We denote $\tau$ as temperature, $\gamma$ as SOC penalty, $\beta_t$ as a noise scheduler, its Riemann integral $\alpha_{s,t} = \int_s^t \beta_z dz$, and $\phi_{s,t} = \exp(-\alpha_{s,t})$. We shorten the notation by $\alpha_t \equiv \alpha_{0,t}$, $\phi_t \equiv \phi_{0,t}$. For original choices of $\beta_t$, $\tau$, or $\gamma$ for each method please see Table 3.

| Methods | Forward SDE $d\mathbf{x}_t = \mathbf{f}(\mathbf{x}_t,t,\mathbf{y})\,dt + g(t)\,d\mathbf{w}_t$ | | Transition kernel $p_t(\mathbf{x}_t\vert\mathbf{x}_0,\mathbf{y}) = \mathcal{N}(\mathbf{x}_t; \boldsymbol{\mu}_t(\mathbf{x}_0,\mathbf{y}), \sigma_t^2\mathbb{I})$ | | Base dist. |
| :---: | :---: | :---: | :---: | :---: | :---: |
| | $\mathbf{f}(\mathbf{x}_t,t,\mathbf{y})$ | $g^2(t)$ | $\boldsymbol{\mu}_t(\mathbf{x}_0,\mathbf{y})$ | $\sigma_t^2$ | $p_1(\mathbf{x}_1\vert\mathbf{y})$ |
| DM-VE | $0$ | $\beta_t$ | $\mathbf{x}_0$ | $\alpha_t$ | $\mathcal{N}(\mathbf{x}_0, \alpha_1\mathbb{I})$ |
| DM-VP | $-\beta_t\mathbf{x}_t$ | $2\beta_t$ | $\phi_t\mathbf{x}_0$ | $1-\phi_t^2$ | $\mathcal{N}(\mathbf{0}, \mathbb{I})$ |
| FM | | $2(1-\phi_t)\beta_t$ | | $(1-\phi_t)^2$ | |
| IR-SDE | $\beta_t(\mathbf{y}-\mathbf{x}_t)$ | $2\tau^2\beta_t$ | $\phi_t\mathbf{x}_0 + (1-\phi_t)\mathbf{y}$ | $\tau^2(1-\phi_t^2)$ | $\mathcal{N}(\mathbf{y}, \tau^2\mathbb{I})$ |
| ResShift | | $\tau^2(2-\phi_t)\beta_t$ | | $\tau^2(1-\phi_t)$ | |
| InDI | | $2\tau^2(1-\phi_t)\beta_t$ | | $\tau^2(1-\phi_t)^2$ | |
| BBDM / DDBM-VE / I²SB | $\frac{\beta_t}{\alpha_{t,1}}(\mathbf{y}-\mathbf{x}_t)$ | $\beta_t$ | $\frac{\alpha_{t,1}}{\alpha_1}\mathbf{x}_0 + \frac{\alpha_t}{\alpha_1}\mathbf{y}$ | $\frac{\alpha_t\alpha_{t,1}}{\alpha_1}$ | $\delta(\mathbf{x}-\mathbf{y})$ |
| DDBM-VP | $\beta_t\big(\frac{2\phi_{t,1}}{1-\phi_{t,1}^2}\mathbf{y} -\frac{1+\phi_{t,1}^2}{1-\phi_{t,1}^2}\mathbf{x}_t\big)$ | $2\beta_t$ | $\phi_t\frac{1-\phi_{t,1}^2}{1-\phi_1^2}\mathbf{x}_0 +\phi_{t,1}\frac{1-\phi_t^2}{1-\phi_1^2}\mathbf{y}$ | $\frac{(1-\phi_{t,1}^2)(1-\phi_t^2)}{1-\phi_1^2}$ | |
| GOUB | $\beta_t\frac{1+\phi_{t,1}^2}{1-\phi_{t,1}^2}(\mathbf{y}-\mathbf{x}_t)$ | $2\tau^2\beta_t$ | $\phi_t\frac{1-\phi_{t,1}^2}{1-\phi_1^2}\mathbf{x}_0 +(1-\phi_t\frac{1-\phi_{t,1}^2}{1-\phi_1^2})\mathbf{y}$ | $\tau^2\frac{(1-\phi_{t,1}^2)(1-\phi_t^2)}{1-\phi_1^2}$ | |
| UniDB† | $\beta_t\frac{(\gamma\tau^2)^{-1}+1+\phi_{t,1}^2}{(\gamma\tau^2)^{-1}+1-\phi_{t,1}^2}(\mathbf{y}-\mathbf{x}_t)$ | | $\phi_t\frac{\gamma^{-1}+1-\phi_{t,1}^2}{\gamma^{-1}+1+\phi_1^2}\mathbf{x}_0 +(1-\phi_t\frac{\gamma^{-1}+1-\phi_{t,1}^2}{\gamma^{-1}+1+\phi_1^2})\mathbf{y}$ | | |

† We consider UniDB that modifies GOUB, which was also used as a main example in the original work.

(Table 15 in Appendix) reveals that performance depends on the task: unconditional processes are favored for super-resolution and low-light enhancement, diffusion bridges are most effective for colorization, and both unconditional and bridge-based methods are competitive for deraining. We then disentangle attributes that impact the differences in performance and discuss consistent trends below.

**Temperature influence.** In this work, we identify temperature as a key factor influencing performance, particularly for Ornstein–Uhlenbeck processes. Figure 5 shows that the reduced performance of InDI is closely tied to its default low-temperature setting ($\tau = 0.06$). The same figure further demonstrates that ResShift ($\tau = 2$) exhibits similarly degraded performance when its temperature is lowered to match that of InDI. This indicates that the observed performance drop is not specific to a particular method, but rather reflects a general limitation of conditional stochastic processes under low-temperature regimes. In particular, reducing the temperature decreases the variance of the

forward marginals, which in turn limits the diversity and quality of the generated samples. We identify two primary mechanisms through which low temperature leads to poorer performance.

First, under low-temperature settings, the intermediate samples $\mathbf{x}_t$ remain tightly concentrated around the line segment connecting the degraded input $\mathbf{y}$ and the target image $\mathbf{x}_0$, resulting in minimal stochasticity during training. Since the network is conditioned on $(\mathbf{x}_t, \mathbf{y}, t)$, it can exploit this near-deterministic structure by learning to extrapolate the displacement $\mathbf{x}_t - \mathbf{y}$ as a simple function of time, rather than modeling the true score of the conditional distribution. Consequently, the model tends to propagate its previous predictions instead of progressively refining them. During inference, where the location of $\mathbf{x}_t$ depends on earlier network outputs, this extrapolation bias compounds over time, leading to diminishing improvements as sampling proceeds. This behavior is illustrated in Figure 4 in Appendix, where low-temperature models exhibit strong collinearity between $\mathbf{x}_t$, $\mathbf{y}$, and the predicted $\hat{\mathbf{x}}_{0\vert t}$, accompanied by slower reduc-

*Table 2.* Results on four selected tasks using ancestral sampling with Network Forward Evaluations (NFE) = 35. The best values are **bolded** and second to best are underscored. We do not notice any significant improvements of conditional processes over standard diffusion. In fact, in super-resolution and low-light image enhancement diffusion achieves best results.

| Model | Facial Super-Resolution | | | Low-light Enhancement | | | Colorization | | | Deraining | | | General Super-Resolution | | |
|---|---|---|---|---|---|---|---|---|---|---|---|---|---|---|---|
| | PSNR | SSIM | LPIPS | PSNR | SSIM | LPIPS | PSNR | SSIM | LPIPS | PSNR | SSIM | LPIPS | PSNR | SSIM | LPIPS |
| DM-VP | **27.86** | **0.748** | **0.181** | 23.01 | 0.685 | **0.186** | 21.73 | 0.685 | 0.191 | 29.02 | 0.853 | 0.055 | 24.12 | 0.663 | 0.381 |
| FM | 27.15 | 0.730 | 0.183 | 22.57 | 0.678 | 0.209 | 21.64 | 0.685 | 0.191 | 28.89 | 0.844 | 0.057 | 23.80 | 0.664 | 0.399 |
| IR-SDE | 26.97 | 0.718 | 0.184 | 23.07 | 0.679 | 0.237 | 22.02 | 0.691 | 0.187 | 28.15 | 0.827 | 0.061 | 23.01 | 0.622 | 0.430 |
| ResShift | 27.66 | 0.746 | 0.184 | 23.15 | **0.693** | 0.195 | 21.82 | 0.689 | 0.189 | 29.17 | 0.856 | 0.052 | **24.36** | **0.671** | **0.378** |
| InDI | 27.21 | 0.713 | 0.199 | 21.48 | 0.633 | 0.323 | 22.28 | 0.699 | 0.191 | 28.17 | 0.816 | 0.069 | 24.20 | 0.664 | 0.384 |
| BBDM | 27.14 | 0.728 | 0.252 | 22.84 | 0.687 | 0.191 | 22.17 | 0.698 | 0.182 | **30.10** | **0.873** | **0.049** | 22.50 | 0.611 | 0.444 |
| DDBM-VE | 27.11 | 0.724 | 0.186 | **23.18** | 0.680 | 0.224 | 22.36 | 0.700 | **0.181** | 28.65 | 0.837 | 0.054 | 23.95 | 0.643 | 0.421 |
| DDBM-VP | 27.46 | 0.741 | 0.190 | 22.55 | 0.659 | 0.231 | 22.23 | 0.697 | 0.184 | 28.94 | 0.842 | 0.052 | 24.07 | 0.641 | 0.419 |
| I$^2$SB | 27.46 | 0.733 | 0.184 | 21.86 | 0.655 | 0.245 | **22.39** | **0.701** | 0.182 | 29.29 | 0.854 | 0.050 | 22.71 | 0.619 | 0.418 |
| GOUB | 27.24 | 0.735 | 0.224 | 22.57 | 0.669 | 0.233 | 21.59 | 0.681 | 0.193 | 28.38 | 0.831 | 0.061 | 23.30 | 0.637 | 0.412 |
| UniDB | 27.59 | 0.739 | **0.181** | 22.80 | 0.679 | 0.223 | 21.69 | 0.684 | 0.190 | 28.02 | 0.824 | 0.060 | 23.24 | 0.633 | 0.417 |

tions in LPIPS.

Second, low-temperature training restricts the model to a narrow region of the state space between the conditioning and target distributions. During inference, even small deviations from this region—caused by modeling inaccuracies or numerical errors—can push the trajectory into areas that are poorly represented in the training data. Because the injected noise is minimal, these deviations are not smoothed out and instead accumulate over time, further degrading reconstruction quality.

Based on these observations, it is clear that diffusion and flow matching do not have this problem because, in the processes definitions, the intermediate sample $x_t$ does not depend on the $y$, as these processes are unconditional. Because $x_t$ and $y$ are independent, network cannot learn to extrapolate and is not biased toward its previous predictions during the inference. Additionally, in this case the forward process diffuses the initial condition $x_0$ almost uniformly in all directions, rather than toward the corresponding $y$. This neglects the second issue with the risk of out-of-distribution trajectories, since the entire space is modeled uniformly.

**Diffusion bridge methods differences.** Diffusion bridges can differ in both their stochastic processes and in how terminal constraints are enforced. UniDB regularizes the forward process by terminating closer to the clean image rather than exactly at the conditioning image $y$, which shifts the learned marginals, leading the network $v_\theta$ to train on shorter trajectories and make smaller, more conservative updates. This regularization reduces overshooting of $x_0$ and keeps trajectories within well-modeled regions of the data manifold. Since UniDB and GOUB share the same stochastic process and differ only in the strength of endpoint enforcement controlled by the parameter $\gamma$, this endpoint regularization explains UniDB's performance gains in almost all tasks in Table 2.

**Samplers and discretization techniques comparison.** In addition, we study several samplers to determine which

work best with our considered methods. The results shown in Tables 13 and 14 in Appendix reveal that ancestral sampling is the most consistent method for all models. The Euler method on reversed ODE works very well for diffusion and OU processes. For all diffusion bridges, deterministic samplers (e.g., Euler ODE, Exponential Integrator ODE, mean-reverting ODE, and second-order Runge-Kutta methods) yield high LPIPS scores ($> 0.3$), suggesting that these methods perform best when combined with stochastic samplers. This is because trajectories start from fixed points and the only source of stochasticity in these models is the Brownian motion, which deterministic samplers omit. As a result, the trajectories follow straight lines between $y$ and $x_0$, converging to the averaged ground truth which is known as the oversmoothing problem.

Finally, we conduct a series of experiments with different discretization techniques. The details can be found in Appendix (techniques plotted in Figure 2 and quantitative results in Table 16). However, we find that the discretization choice has no significant impact on final sample quality.

**Additional results** Appendix G includes additional experiments that are not part of the main analysis of the impact of process definition on sample quality. Specifically, we report results obtained using a different backbone architecture (DiT instead of U-Net), as well as results using the original parameter settings of the compared methods.

## 4. Conclusion

In this work, we propose a unified continuous-time framework for stochastic image enhancement that brings together a wide range of existing methods under a common stochastic differential equation formulation. By classifying popular approaches as instances of standard diffusion processes, Ornstein–Uhlenbeck processes, or diffusion bridges, we disentangle core model definitions from schedulers, samplers, and discretization choices, enabling fair and transparent comparison. This perspective reveals that many methods previously

treated as distinct differ primarily in design choices such as endpoint conditioning, temperature, and sampling strategy rather than in their underlying stochastic processes. Leveraging the unified framework, we conduct a controlled empirical study across multiple image enhancement tasks and find no consistently dominant method. Instead, we identify the key factors that govern performance differences and explain observed empirical trends. To support reproducibility and future research, we release ItoVision, a modular library that directly implements the unified formulation and facilitates systematic development of new methods.

## Impact Statement

Our work unifies definitions of existing methods and evaluates them on standard benchmarks. We do not introduce new datasets; therefore, we do not find any ethical concerns associated with this study.

## Acknowledgments

The work conducted by Wojciech Kozłowski, Radosław Kuczbański and Maciej Zięba was supported by the National Centre of Science (Poland) grant no. 2021/43/B/ST6/02853.

Calculations have been carried out in Wroclaw Centre for Networking and Supercomputing (http://www.wcss.pl), grant No. 1766404231.

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

# Unifying Deep Stochastic Processes for Image Enhancement
## Appendix

## A. Proofs and methods derivations

### A.1. Flow Matching Diffusion

**Proposition 2.1.** *Let $\mathbf{x}_t$ be a continuous stochastic process that follows the SDE*

$$\mathrm{d}\mathbf{x}_t = -\beta_t \mathbf{x}_t\, \mathrm{d}t + \sqrt{2(1-\phi_t)\beta_t}\, \mathrm{d}\mathbf{w}_t. \tag{4}$$

*If the scheduler is defined as $\beta_t = \frac{1}{1-t}$, then for any $\mathbf{x}_0 \sim p_{data}(\mathbf{x}_0)$ and $\epsilon \sim \mathcal{N}(\mathbf{0}, \mathbb{I})$ we have $\mathbf{x}_t = (1-t)\mathbf{x}_0 + t\epsilon$ for $t \in [0, 1)$, which corresponds to transitions of (Liu et al., 2022).*

*Proof.* To construct the solution, we take the Brownian motion in the interval $[0, 1]$ and consider the modified version of the (4) in the closed interval $[0, 1]$ where $\beta_t$ is replaced by $\tilde{\beta}_t$ defined as follows: $\tilde{\beta}_t = \beta_t$, for $t \in [0, 1-\delta]$ and $\tilde{\beta}_t = \beta_{1-\delta}$, for $t \in [1-\delta, 1]$, for a fixed $\delta \in (0, 1)$. The modified equation has a unique strong solution (see (Øksendal, 2003) Theory 5.2.1 or (Karatzas & Shreve, 2014) Theory 5.2.5 and 5.5.2.9). If we take $\delta, \delta' \in (0, 1)$ and $\delta < \delta'$, due to pathwise uniqueness, almost all trajectories of the corresponding solutions $\mathbf{x}_t^\delta, \mathbf{x}_t^{\delta'}$ agree on $[0, \delta']$. Therefore, $\mathbf{x}_t^\delta$ is an extension of the solution $\mathbf{x}_t^{\delta'}$ of (4) from the interval $[0, 1-\delta']$ to $[0, 1-\delta]$. The solution can now be constructed for almost all trajectories by taking $\delta \to 0^+$. Similarly, by truncating the interval $[0, 1)$ to $[0, 1-\delta]$ one proves the pathwise uniqueness.

We find the solution of (4) using Integrating Factor method. First, denote that the integrating factor $M$ of (4) is equivalent to the definition of $\phi_{s,t}$

$$M = \exp\left(\int_s^t a(z)dz\right) = \phi_{s,t} = \frac{1-t}{1-s}. \tag{19}$$

The mild solution is therefore

$$\mathbf{x}_t = \phi_t \mathbf{x}_0 + \int_0^t \phi_{s,t}g(s)\mathrm{d}\mathbf{w}_s \tag{20}$$

Now, for simplicity, we divide our calculation into expected value $\mathbb{E}[\mathbf{x}_t|\mathbf{x}_0]$ and variance $\mathrm{Var}(\mathbf{x}_t|\mathbf{x}_0)$ of transition densities. Because the Ito integral has zero mean, only the deterministic part of (20) contributes to the expected value

$$\mathbb{E}[\mathbf{x}_t|\mathbf{x}_0] = \phi_t \mathbf{x}_0 \tag{21}$$

Similarly, for variance, we consider only the stochastic part of (20).

$$\mathrm{Var}(\mathbf{x}_t) = \mathbb{E}\left[\left(\int_0^t \phi_{s,t}g(s)\mathrm{d}\mathbf{w}_s\right)^2\right] \tag{22}$$

We can apply Ito isometry to get

$$\mathrm{Var}(\mathbf{x}_t) = \int_0^t \phi_{s,t}^2 g^2(s)ds. \tag{23}$$

Using exponential properties and linearity of Riemann integral we have

$$\phi_{s,t} = \frac{\phi_t}{\phi_s}, \tag{24}$$

we can apply that to (23) to obtain

$$\mathrm{Var}(\mathbf{x}_t) = \phi_t^2 \int_0^t \phi_s^{-2} g^2(s)ds. \tag{25}$$

Recall that $g^2(t) = 2(1-\phi_t)\beta_t$ which gives

$$\mathrm{Var}(\mathbf{x}_t) = 2\phi_t^2 \int_0^t \phi_s^{-2}\beta_s(1-\phi_s)ds. \tag{26}$$

We substitute $u = \phi_s^{-1}, du = \phi_s^{-1}\beta_s ds$

$$\begin{aligned}
\mathrm{Var}(\mathbf{x}_t) &= 2\phi_t^2 \int_1^{\phi_t^{-1}} u - 1 \, du \\
&= 2\phi_t^2 \left(\frac{\phi_t^{-2}}{2} - \phi_t^{-1} + \frac{1}{2}\right) \\
&= 1 - 2\phi_t + \phi_t^2 \\
&= (1-\phi_t)^2 \tag{27}
\end{aligned}$$

Recall that for our chosen scheduler, we have $\phi_t = 1 - t$, therefore

$$\mathbb{E}[\mathbf{x}_t|\mathbf{x}_0] = (1-t)\mathbf{x}_0 \tag{28}$$
$$\mathrm{Var}(\mathbf{x}_t) = t^2 \tag{29}$$

and we can express intermediate sample $\mathbf{x}_t$ as

$$\mathbf{x}_t = (1-t)\mathbf{x}_0 + t\epsilon \tag{30}$$

for some $\mathbf{x}_0 \sim p_{\text{data}}(\mathbf{x}_0)$ and $\epsilon \sim \mathcal{N}(\mathbf{0}, \mathbb{I})$. That shows that (4) has the same marginals as (Liu et al., 2022).  □

### A.2. ResShift

**Proposition 2.2.** *The Ornstein–Uhlenbeck process defined by the stochastic differential equation*

$$\mathrm{d}\mathbf{x}_t = \beta_t(\mathbf{y} - \mathbf{x}_t)\, \mathrm{d}t + \tau\sqrt{\beta_t(2-\phi_t)}\, \mathrm{d}\mathbf{w}_t \tag{11}$$

*induces the same transition densities as the original ResShift model in Eq. (10) for $t \in [0, 1]$.*

*Proof.* Based on Theory 5.2.1 from (Øksendal, 2003), (11) has a strong solution and is unique.

Similarly to Proposition 2.1, we can find the solution using the integrating factor method; the subsequent steps are very similar, so we omit some comments.

$$\mathbf{x}_t = \phi_t \mathbf{x}_0 + \mathbf{y} \int_0^t \phi_{s,t} \beta_s ds + \int_0^t \phi_{s,t} g(s) \mathrm{d}\mathbf{w}_s \quad (31)$$

$$\begin{aligned}
\mathbb{E}[\mathbf{x}_t | \mathbf{x}_0, \mathbf{y}] &= \phi_t \mathbf{x}_0 + \mathbf{y} \int_0^t \phi_{s,t} \beta_s ds \\
&= \phi_t \mathbf{x}_0 + \phi_t \mathbf{y} \int_0^t \phi_s^{-1} \beta_s ds \\
&= \phi_t \mathbf{x}_0 + \phi_t \mathbf{y} \int_1^{\phi_t^{-1}} dr \\
&= \phi_t \mathbf{x}_0 + (1 - \phi_t)\mathbf{y} \quad (32)
\end{aligned}$$

$$\begin{aligned}
\mathrm{Var}(\mathbf{x}_t) &= \mathbb{E}\left[\left(\int_0^t \phi_{s,t} g(s) \mathrm{d}\mathbf{w}_s\right)^2\right] \\
&= \int_0^t \phi_{s,t}^2 g^2(s) ds \\
&= \tau^2 \phi_t^2 \int_0^t \phi_s^{-2} \beta_s (2 - \phi_s) ds \\
&= \tau^2 \phi_t^2 \int_1^{\phi_t^{-1}} 2r - 1 \, dr \\
&= \tau^2 \phi_t^2 \left(\phi_t^{-2} - \phi_t^{-1}\right) \\
&= \tau^2 \left(1 - \phi_t\right) \quad (33)
\end{aligned}$$

Combining expected value and variance, we get the transition density formula from (12). □

### A.3. InDI

**Proposition 2.3.** *The OU-process with the following stochastic differential equation:*

$$\mathrm{d}\mathbf{x}_t = \beta_t(\mathbf{y} - \mathbf{x}_t) \, \mathrm{d}t + \tau\sqrt{2\beta_t(1 - \phi_t)} \, \mathrm{d}\mathbf{w}_t. \quad (15)$$

*If the scheduler is defined as $\beta_t = \frac{1}{1-t}$, then for any $\mathbf{x}_0 \sim p_{data}(\mathbf{x}_0)$ and $\epsilon \sim \mathcal{N}(\mathbf{0}, \mathbb{I})$ we have the same marginals as* (14) *for $t \in [0, 1)$.*

*Proof.* The conditions are the same as in 2.1.

Similarly to Proposition 2.1, we can find the solution using the integrating factor method; the subsequent steps are very similar, so we omit some comments.

$$\mathbf{x}_t = \phi_t \mathbf{x}_0 + \mathbf{y} \int_0^t \phi_{s,t} \beta_s ds + \int_0^t \tau\phi_{s,t}\sqrt{2\beta_s(1 - \phi_s)}\mathrm{d}\mathbf{w}_s \quad (34)$$

The expected value is the same as in Proposition 2.2. The variance is a scaled version of the variance from the Proposition 2.1. Therefore, we have

$$\mathbb{E}[\mathbf{x}_t | \mathbf{x}_0, \mathbf{y}] = \phi_t \mathbf{x}_0 + (1 - \phi_t)\mathbf{y} \quad (35)$$

$$\mathrm{Var}(\mathbf{x}_t) = \tau^2 (1 - \phi_t)^2 \quad (36)$$

If we set $\phi_t = 1 - t$ then we get

$$p_t(\mathbf{x}_t | \mathbf{x}_0, \mathbf{y}) = \mathcal{N}(\mathbf{x}_t; (1 - t)\mathbf{x}_0 + t\mathbf{y}, \tau^2 t^2 \mathbb{I}), \quad (37)$$

which is equivalent to (14). □

### A.4. I²SB

**Remark 2.5.** Deriving the bridge process from (Liu et al., 2023) Theory 3.1 with $\mathbf{f}(\mathbf{x}_t, t) = \mathbf{0}$ and $g(t) = \sqrt{\beta_t}$ leads to the same result as applying h-transform (Doob, 1984) to the exact same SDE $\mathrm{d}\mathbf{x}_t = \sqrt{\beta_t}\,\mathrm{d}\mathbf{w}_t$. This gives the definition of DDBM-VE (Zhou et al., 2023b).

We consider two SDEs where, respectively, time goes forward and backward

$$\mathrm{d}\mathbf{x}_t = \left[\mathbf{f}(\mathbf{x}_t, t) + g^2(t)\nabla_{\mathbf{x}_t} \log \Psi(\mathbf{x}_t, t)\right] \mathrm{d}t + g(t)\,\mathrm{d}\mathbf{w}_t, \quad (38)$$

$$\mathrm{d}\mathbf{x}_t = \left[\mathbf{f}(\mathbf{x}_t, t) - g^2(t)\nabla_{\mathbf{x}_t} \log \hat{\Psi}(\mathbf{x}_t, t)\right] \mathrm{d}t + g(t)\,\mathrm{d}\bar{\mathbf{w}}_t, \quad (39)$$

$\mathbf{x}_0 \sim p_{data}(\mathbf{x}_0)$, $\mathbf{x}_1 \sim p_{low}(\mathbf{x}_1)$, and the functions $\Psi$ and $\hat{\Psi}$ are the solution to the following coupled PDEs[3]

$$\begin{cases}
\dfrac{\partial \Psi}{\partial t} = -\nabla_{\mathbf{x}}\Psi^\mathsf{T}\mathbf{f} - \dfrac{1}{2}\mathrm{Tr}(g^2\nabla_{\mathbf{x}}^2\Psi), \\
\dfrac{\partial \hat{\Psi}}{\partial t} = -\nabla_{\mathbf{x}} \cdot (\hat{\Psi}\mathbf{f}) + \dfrac{1}{2}\mathrm{Tr}(g^2\nabla_{\mathbf{x}}^2\hat{\Psi}),
\end{cases} \quad (40)$$

$$\begin{aligned}
s.t. \quad &\Psi(\mathbf{x}, 0)\hat{\Psi}(\mathbf{x}, 0) = p_{data}(\mathbf{x}), \\
&\Psi(\mathbf{x}, 1)\hat{\Psi}(\mathbf{x}, 1) = p_{low}(\mathbf{x}). \quad (41)
\end{aligned}$$

From Theory 3.1 from (Liu et al., 2023) when the above system holds, we can consider $\hat{\Psi}(\cdot, 0), \Psi(\cdot, 1)$ as the boundary distributions and $\nabla_{\mathbf{x}_t} \log \hat{\Psi}(\mathbf{x}_t, t)$ and $\nabla_{\mathbf{x}_t} \log \Psi(\mathbf{x}_t, t)$ are the score functions for the following SDEs, respectively

$$\mathrm{d}\mathbf{x}_t = \mathbf{f}(\mathbf{x}_t, t)\,\mathrm{d}t + g(t)\,\mathrm{d}\mathbf{w}_t, \quad \mathbf{x}_0 \sim \hat{\Psi}(\mathbf{x}_0, 0) \quad (42)$$

$$\mathrm{d}\mathbf{x}_t = \mathbf{f}(\mathbf{x}_t, t)\,\mathrm{d}t + g(t)\,\mathrm{d}\bar{\mathbf{w}}_t, \quad \mathbf{x}_1 \sim \Psi(\mathbf{x}_1, 1), \quad (43)$$

---

[3]Following (Chen et al., 2021), for brevity we denote $\Psi \equiv \Psi(\mathbf{x}, t), \mathbf{f} \equiv \mathbf{f}(\mathbf{x}, t)$, and $g \equiv g(t)$.

with the same $\mathbf{f}$ and $g$ as in (38) and (39). For simplicity, we specify the drift $\mathbf{f}(\mathbf{x}_t, t) = \mathbf{0}$ and the diffusion coefficient $g(t) = \sqrt{\beta_t}$ because that was the practical design choice made in (Liu et al., 2023). With that we can easily calculate the transition kernel for (43) remembering that time goes backwards

$$p_t(\mathbf{x}_t|\mathbf{x}_1) = \mathcal{N}\left(\mathbf{x}_t; \mathbf{x}_1, \int_t^1 \beta_s ds \mathbb{I}\right). \quad (44)$$

To remain consistent with our notation, we denote $\alpha_{t,1} = \int_t^1 \beta_s ds$. With that, we can define the score of (43) as $\nabla_{x_t} \log \Psi(\mathbf{x}_t, t) = \frac{\mathbf{x}_1 - \mathbf{x}_t}{\alpha_{t,1}}$. Finally, we substitute the definitions of $\mathbf{f}$, $g$, and $\nabla_{x_t} \log \Psi(\mathbf{x}_t, t)$ into (38) which yields

$$d\mathbf{x}_t = \frac{\beta_t}{\alpha_{t,1}}(\mathbf{x}_1 - \mathbf{x}_t)\, dt + \sqrt{\beta_t}\, d\mathbf{w}_t. \quad (45)$$

To further unify the notation, we denote $\mathbf{y} \equiv \mathbf{x}_1$, which is true for all the models based on bridges that we consider

$$d\mathbf{x}_t = \frac{\beta_t}{\alpha_{t,1}}(\mathbf{y} - \mathbf{x}_t)\, dt + \sqrt{\beta_t}\, d\mathbf{w}_t. \quad (46)$$

This corresponds to the h-transform (Doob, 1984) applied to variance exploding diffusion process (Zhou et al., 2023b; Li et al., 2023).

## B. Unified schedulers

In this section, we show all the schedulers that were used in the methods we covered. Each scheduler is defined for $t \in [0, 1]$.

We provide the exact definitions for $\beta_t$ and $\alpha_{s,t} = \int_s^t \beta_z dz$. The $\phi_{s,t} = \exp(-\alpha_{s,t})$ can be easily computed for any scheduler.

**Linear** with two parameters $\beta_{min}$, and $\beta_{max}$ that denote the value of $\beta_t$ at time 0, and 1.

$$\beta_t = (\beta_{max} - \beta_{min})t + \beta_{min} \quad (47)$$

$$\alpha_{s,t} = \frac{1}{2}(\beta_{max} - \beta_{min})(t^2 - s^2) + \beta_{min}(t - s) \quad (48)$$

**Cosine** was used among several papers with two parameters $\epsilon$ and $\delta$ that are used to ensure numerical stability. For all methods, we have

$$\epsilon = 0.008, \delta = 0.005.$$

For clarity, we shall define a few intermediate functions

$$g(t) = \cos\left(\frac{t + \epsilon}{1 + \epsilon} \cdot \frac{\pi}{2}\right)^2 \quad (49)$$

$$h(t) = \sin\left(\frac{t + \epsilon}{1 + \epsilon}\pi\right) \quad (50)$$

$$f(t) = 1 - \frac{g(t)}{g(0)} \quad (51)$$

In previous works (Luo et al., 2023c; Yue et al., 2023a; Zhu et al., 2025), the integral of the function $f(t)$ was calculated numerically. This approach limits both speed and flexibility, as the values of $\int_0^{t_i} f(s)ds$ must be recomputed for each discrete timestep $\{t_i\}_{i=0}^K$. In contrast, we derive a closed-form analytical solution, given by

$$F(s, t) = \int_s^t f(z)dz = t - s +$$
$$\frac{(s - t)\pi + (1 + \epsilon)(h(s) - h(t))}{2\pi g(0)}. \quad (52)$$

With that, we can easily define the scheduler

$$\beta_t = -\log(\delta)\frac{f(t)}{F(0, 1)} \quad (53)$$

$$\alpha_{s,t} = -\log(\delta)\frac{F(s, t)}{F(0, 1)} \quad (54)$$

**Exponential** with three parameters $\eta_{min}$, $\eta_{max}$, and $p$, where $\eta_{min}$ and $\eta_{max}$ are responsible for a value range of $\phi$, and $p$ influence how fast $\phi$ goes to $\eta_{max}$.

$$\beta_t = 2p \log\left(\frac{\eta_{max}}{\eta_{min}}\right) t^{p-1} \eta_{min}^2 \left(\frac{\eta_{max}}{\eta_{min}}\right)^{2t^p} \phi_t \quad (55)$$

$$\alpha_{s,t} = \log\left(\frac{1 - \left(\eta_{min}^2\left(\frac{\eta_{max}}{\eta_{min}}\right)^{2s^p}\right)}{1 - \left(\eta_{min}^2\left(\frac{\eta_{max}}{\eta_{min}}\right)^{2t^p}\right)}\right) \quad (56)$$

**Inversed** is defined so that $\phi_t = 1 - t$ and it is a useful scheduler for methods based on linear interpolation such as Flow Matching or InDI.

$$\beta_t = \frac{1}{1 - t} \quad (57)$$

$$\alpha_{s,t} = \log\left(\frac{1 - s}{1 - t}\right) \quad (58)$$

**Quadratic Symmetric** is used when we want the transition variance of a diffusion bridge to be symmetric at $t \in [0, 1]$.

$$\beta_t = \left(\left(\sqrt{\beta_{max}} - \sqrt{\beta_{min}}\right)\left(\frac{1}{2} - \left|\frac{1}{2} - t\right|\right) + \sqrt{\beta_{min}}\right)^2 \quad (59)$$

We consider two scenarios $t \leq 0.5$ and $t > 0.5$.

$$f(t)^{(1)} = \left( \sqrt{\beta_{max}} - \sqrt{\beta_{min}} \right)^2 \frac{t^3}{3} +$$
$$\left( \sqrt{\beta_{max}} - \sqrt{\beta_{min}} \right) \sqrt{\beta_{min}} t^2 + \beta_{min} t \quad (60)$$

$$f(t)^{(2)} = \left( \sqrt{\beta_{max}} - \sqrt{\beta_{min}} \right)^2 \cdot \left( t - t^2 + \frac{t^3}{3} \right) +$$
$$\left( \sqrt{\beta_{max}} - \sqrt{\beta_{min}} \right) \sqrt{\beta_{min}} \cdot (2t - t^2) + \beta_{min} t$$
$$(61)$$

$$\alpha_t = \begin{cases} f(t)^{(1)}, & t \leq \frac{1}{2} \\ f(t)^{(2)} - f(\frac{1}{2})^{(2)} + f(\frac{1}{2})^{(1)}, & t > \frac{1}{2} \end{cases} \quad (62)$$

$$\alpha_{s,t} = \alpha_t - \alpha_s \quad (63)$$

## C. Library

*Ito Vision* is a Python library, built on PyTorch, that implements 11 considered methods, together with multiple schedulers, samplers, network parametrizations and discretization techniques. Collectively, these components support the use of all methods presented in this paper and enable straightforward implementation of novel approaches. The library can be installed via:

```
$ pip install <path/to/ito_vision>
```

The codebase is fully type-hinted and designed for consistency through the use of abstract classes for each module. For example, every method inherits from the `IterativeRefinementMethod` abstract class, which specifies required functions that correspond to the definitions from Table 1, bridging the gap between the mathematical formulations and the code, and enabling easy implementation of new stochastic processes.

Further details and a code example are provided in the library's `README.md`, included in the supplementary material.

## D. Implementation details

Below we provide detailed description of the experimental setups for each task. We evaluate the methods using Peak Signal-to-Noise Ratio (PSNR), Structural Similarity Index Measure (SSIM), Learned Perceptual Image Patch Similarity (LPIPS) (Zhang et al., 2018), and Natural Image Quality Evaluator (NIQE) (Mittal et al., 2012). For image super-resolution on FFHQ and colorization on ImageNet, we also report the Fréchet Inception Distance (FID) (Heusel et al.,

2017), as the test datasets are large enough for this measure. We use LPIPS as the main metric to evaluate model quality because, unlike PSNR and SSIM, it aligns with human perception, and unlike NIQE and FID, it incorporates ground truth, which is important for preserving details.

**Facial Image Super-Resolution** For this task, we used the FFHQ dataset (Karras et al., 2019) and performed x8 single image super-resolution $64 \times 64 \rightarrow 512 \times 512$. In addition to reducing resolution, we also applied JPEG compression to 10% quality and then upscaled back to $512 \times 512$. Both resizing used bicubic interpolation with anti-aliasing. Each model was trained in $256 \times 256$ crops with batch size of 64. We split the data set that contains 70k images into train-val-test splits with $0.98 : 0.002 : 0.018$ proportions. For our backbone, we used UNet (Ronneberger et al., 2015) with 119M parameters from the diffuser library (von Platen et al., 2022) with the same channel dimensionality as in (Dhariwal & Nichol, 2021) (see their Appendix I, Table 12 ImageNet $256 \times 256$). Detailed architectures are shown in Table 4. Each method was trained for 400k iterations. The final model was selected based on the LPIPS metric (Zhang et al., 2018) on the validation split, measured each 10k iterations.

**Low-Light Image Enhancement** We used the LOL dataset (Wei et al., 2018), as it contains a relatively extensive collection of 500 real-world dark and light pairs of the same scene. We used original train and test splits, and we moved 5 pairs from training to validation split as it was not provided. We trained the models on $320 \times 320$ crops with batch size of 80. The model had 31.3M parameters and its architecture was the same as in the super-resolution task, with reduced channel dimensions. Each method was trained for 150k iterations with the validation each 2.5k iterations. Following most of the state-of-the-art methods in this field (Zhang et al., 2019; Cai et al., 2023; Zhou et al., 2023a) we involve conditioning on ground-truth average lightness. We injected it through the attention mechanism and adjusted the value channel for each estimation of ground truth $\hat{x}_{0|t}$ to the given lightness through the HSV color space.

**Image Colorization** Here, we utilized the ImageNet dataset (Russakovsky et al., 2015). It contains a wide range of real world objects; therefore, it fits this task perfectly as the diversity of objects and their hues makes it well-suited for assessing generalization of validated methods. For training, we used all images from the train split, with width and height between 256 and 768. We treat low-quality input $y$ as the grayscale version of ground truth with three channels to maintain the same dimensionality. We used pretrained VAE from Stable Diffusion 2.1 (Rombach et al., 2022) originally used for $\times 4$ super-resolution. We use the same architecture of the UNet as in the image super-resolution task. We train

*Table 3.* Original choices of used scheduler, sampler, and other method-specific parameters.

| Method | Scheduler $\beta_t$ | Sampler | Parametrization | Loss | Parameters |
|---|---|---|---|---|---|
| DM-VP† | linear | Ancestral sampling | $\epsilon_t$ | $l_2$ norm | - |
| FM | inversed | Euler-ODE | $\mathbf{x}_0 - \mathbf{y}$ | $l_2$ norm | - |
| IR-SDE | cosine | Euler-Maruyama | $\epsilon_t$ | $l_2$ norm | $\tau = 0.20$ |
| ResShift | exponential | Ancestral sampling | $\mathbf{x}_0$ | $l_2$ norm | $\tau = 2.00$ |
| InDI | inversed | Euler-ODE | $\mathbf{x}_0$ | $l_1$ norm | $\tau = 0.06$ |
| BBDM | constant | Ancestral sampling | $\epsilon_t$ | $l_2$ norm | - |
| DDBM-VE | linear | 2nd Heun & Langevin-Heun | karras | $l_2$ norm | - |
| DDBM-VP | linear | 2nd Heun & Langevin-Heun | karras | $l_2$ norm | - |
| I$^2$SB | quadratic-symmetric | Ancestral sampling | $\epsilon_t$ | $l_2$ norm | - |
| GOUB | cosine | Euler-ODE & mean-ODE | $\epsilon_t$ | $l_1$ norm | $\tau = 0.34$ |
| UniDB-GOUB | cosine | Euler-ODE & mean-ODE | $\epsilon_t$ | $l_1$ norm | $\tau = 0.34, \gamma = 1e4$ |

† For diffusion we consider DDPM (Ho et al., 2020) implementation.

*Table 4.* Detailed parameters choices of training procedure for each image enhancement task.

| | Facial Super-Resolution | Colorization | Low-light Enhancement | Deraining | General Super-Resolution |
|---|---|---|---|---|---|
| Dataset | FFHQ | ImageNet | LOL | Rain100H | DIV2K |
| Iterations | 400k | 400k | 150k | 50k | 250k |
| Latent | ✗ | ✓ | ✗ | ✗ | ✓ |
| Batch size | 64 | 256 | 80 | 80 | 32 |
| Random crop size | $256 \times 256$ | $384 \times 384$ | $320 \times 320$ | $320 \times 320$ | $512 \times 512$ |
| Learning rate† | 1e-4 → 1e-7 | 1e-4 → 1e-7 | 1e-4 → 1e-7 | 1e-4 → 1e-7 | 1e-4 → 1e-7 |
| Backbone parameters | 119M | 119M | 31.3M | 31.3M | 119M |
| Channels | 128 | 128 | 64 | 64 | 128 |
| Depth | 2 | 2 | 2 | 2 | 2 |
| Channel multipliers | 1,1,2,2,4,4 | 1,1,2,2,4,4 | 1,1,2,2,4,4 | 1,1,2,2,4,4 | 1,1,2,2,4,4 |
| Attention head dimension | 64 | 64 | 64 | 64 | 64 |

† Decreasing learning rate with respect to cosine annealing LR scheduler.

it for $400k$ iteration with $384 \times 384$ random crops and a batch size of $256$. We evaluated FID on $50k$ images from the test split.

**Image Deraining** For the fourth task, we used the Rain1400 dataset (Fu et al., 2017) with image pairs prepared with synthetic rain applied to low-quality images $\mathbf{y}$. Similarly to other experiments, we report PSNR and SSIM metrics in RGB space.

**General Image super-resolution** Here, we used DIV2K (Agustsson & Timofte, 2017) dataset with $\times 4$ degradation, generated using an interpolation method unknown to the model. For this task, we utilized latent diffusion on pre-trained VAE from Stable Diffusion 2.1 (Rombach et al., 2022). We trained the UNet of the same architecture as in Facial ISR for 250k iterations with batch size of 32 and $128 \times 128$ latent crops, which corresponds to $512 \times 512$ training crops in pixel space.

**Transformer backbone** To study the impact of the backbone architecture, we trained each method for image super-resolution on the FFHQ (Karras et al., 2019) dataset using DiT (Peebles & Xie, 2022) as the backbone. All dataset parameters were kept identical to those used in the Facial Image Super-Resolution setup. The DiT model contains 114 M parameters, chosen to approximately match the size of the UNet backbone. It consists of 12 transformer layers with 14 attention heads per layer, where each head has a dimensionality of 48. The model operates on non-overlapping patches of size $8 \times 8$. Although smaller patch sizes would most likely improve performance, we found that this con-

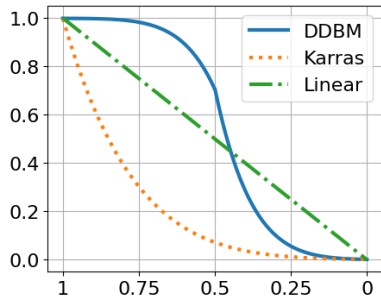

*Figure 2.* Visualization of different discretization techniques. Karras discretization spends most of the time at low values of $t$, while DDBM places emphasis on both the beginning and the end of the trajectory.

figuration closely matches the UNet backbone in terms of computational speed.

**Original setup** To examine the behavior of each method, we conducted experiments on the FFHQ dataset (Karras et al., 2019) using the same training protocol as in the Facial Image Super-Resolution setup. For each method, we keep all parameters at their original values as specified in the original paper, including network parameterization, sampler, loss function, and conditioning strategy. This ensures that our comparison evaluates the methods as complete systems rather than just the processes definition.

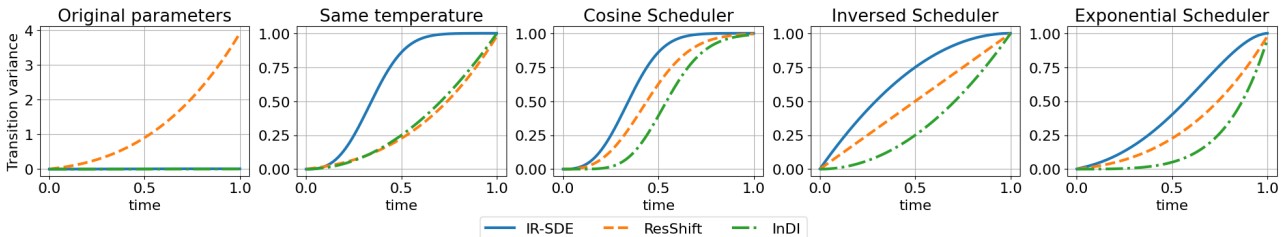

*Figure 3.* Transition variance $\text{Var}(\mathbf{x}_t | \mathbf{x}_0)$ for all considered Ornstein-Uhlenbeck processes. Left: Original temperatures $\tau$ and schedulers $\beta_t$. Next: standarized $\tau$ with original schedulers. Middle to right: standarized $\tau$ with specific schedulers. We can see that ResShift has by far the highest temperature but IR-SDE reaches maximum level of noise faster than other methods. InDI has the lowest temperature and adds the noise with the slowest pace.

## E. The Use of Large Language Models

We used LLMs to find better synonyms and correct grammar in our text to improve its readability. We read, analyzed, and modified each LLM suggestion, if needed, to ensure that there were no hallucinations that could lead to misinformation.

## F. Related works

**SP for image enhancement** Diffusion models were first introduced for unconditional generation (Sohl-Dickstein et al., 2015; Song & Ermon, 2019; Ho et al., 2020; Song et al., 2020b) and later adapted to image enhancement by concatenating the corrupted image with the input (Saharia et al., 2022a;b; Rombach et al., 2022; Jinhui et al., 2023). Another line of work (Wang et al., 2022; Yang et al., 2023) applies guidance to recover a clean version of the low-quality image. Following (Bansal et al., 2023), which showed that the forward process does not have to be Gaussian, the researchers proposed alternative formulations that directly incorporate input into the diffusion process. Examples include ResShift (Yue et al., 2023b), which defines a Markov chain converging to a Gaussian centered on the low-quality image; InDI (Delbracio & Milanfar, 2023), which adapts flow matching (Lipman et al., 2022; Liu et al., 2022) using the low-quality image distribution as the base; IR-SDE (Luo et al., 2023a), which introduces a mean-reverting Ornstein–Uhlenbeck process; I$^2$SB (Liu et al., 2023), derived from the Schrödinger Bridge theory (Chen et al., 2021; De Bortoli et al., 2021; 2024); DDBM and GOUB (Zhou et al., 2023b; Yue et al., 2023a), based on Doob's h-transform (Doob, 1984); and UniDB (Zhu et al., 2025), which is based on stochastic optimal control (SOC). The goal of this paper is to provide a unified formulation for these methods.

**Methods not included in our studies** We exclude methods that use the definition of the degradation function (Kawar et al., 2021; 2022; Chung et al., 2022; Song et al., 2023). We do not consider methods that improve the architectures or samplers of our chosen methods, such as (Hu et al., 2025),

Refusion (Luo et al., 2023b) that focuses on the design of the backbone for the IR-SDE method or I$^3$SB (Wang et al., 2025) that applies DDIM (Song et al., 2020a) sampling to I$^2$SB. Finally, we do not include the Schrödinger Bridge methods (De Bortoli et al., 2021; 2024; Su et al., 2022; Kim et al., 2023), as they are designed for unpaired data, while we assume paired training data.

**Unification** We were inspired by EDM (Karras et al., 2022), which unifies and compares diffusion models within a consistent framework. (Albergo et al., 2025) introduced a common formulation for diffusion- and flow matching based methods. In a similar direction, (Tong et al., 2023) combines the flow-based and Schrödinger Bridge methods for unpaired image translation. For GANs, (Lucic et al., 2018) conducted large-scale studies and demonstrated that different architectures achieve comparable results when trained under the same conditions. In image enhancement, (Li et al., 2025) provided a comprehensive evaluation of diffusion-based methods using their original backbones. We want to fill this gap by unifying and comparing alternative stochastic processes for image enhancement in the same training setup.

## G. Additional results

**Transformer backbone** The results in Table 11 indicate that the relative performance of the methods remains consistent with the experiments using UNet as the backbone. In particular, ResShift achieves the best overall performance, diffusion models continue to produce reasonable results, and InDI performs poorly, especially at low numbers of function evaluations (NFE). Notably, IR-SDE shows a slight relative improvement when using the DiT backbone compared to UNet.

When comparing backbone architectures, we observe that DiT generally underperforms UNet on average. Only ResShift and GOUB achieve better results with DiT than with UNet. Moreover, in contrast to the UNet-based experiments, the performance with DiT degrades as the NFE increases.

**Original setup** The results in Table 12 show that most methods achieve slightly worse performance than in our unified setup. For example, the diffusion model performs worse because predicting $\mathbf{x}_0$ is more effective than predicting epsilon for image enhancement tasks when the network is conditioned on the input. Next, I$^2$SB does not condition the backbone network and relies only on information from $\mathbf{y}$ through the starting point of the diffusion process. As a result, its performance is notably worse, particularly at low numbers of function evaluations (NFE). By using specific samplers, DDBM-VE and DDBM-VP focus more on pixel fidelity (as indicated by high PSNR and SSIM values), while sacrificing some perceptual realism (measured by LPIPS and FID). This setup can work well for relatively simple image enhancement tasks such as colorization or image de-raining. However, it is less suitable for super-resolution with a high upscaling ratio, where the model must hallucinate fine details. Some methods, such as Flow Matching and InDI, improve over their standardized counterparts, suggesting that specific choices of reparameterization, samplers, or loss functions can work particularly well together. This further supports the main observation that, when trained under the same protocol, all methods achieve broadly similar performance.

*Table 5.* Image super-resolution on FFHQ dataset using ancestral sampling with Network forward evaluations (NFE) = 5, 35, and 100. The best values are **bolded** and second to best are underscored.

| Model | NFE = 5 | | | | | NFE = 35 | | | | | NFE = 100 | | | | |
|---|---|---|---|---|---|---|---|---|---|---|---|---|---|---|---|
| | PSNR | SSIM | LPIPS | NIQE | FID | PSNR | SSIM | LPIPS | NIQE | FID | PSNR | SSIM | LPIPS | NIQE | FID |
| DM-VP | **29.04** | **0.7825** | 0.229 | 8.728 | 17.857 | **27.86** | **0.7481** | **0.181** | 7.979 | 17.525 | **27.23** | 0.7217 | **0.164** | 7.652 | 17.760 |
| FM | 28.72 | 0.7756 | 0.241 | 8.818 | 17.565 | 27.15 | 0.7295 | 0.183 | 7.839 | 17.745 | 26.48 | 0.6989 | 0.172 | 7.561 | 17.843 |
| IR-SDE | 28.64 | 0.7712 | 0.249 | 8.974 | 18.230 | 26.97 | 0.7179 | 0.184 | **7.805** | 18.302 | 26.17 | 0.6881 | 0.189 | 7.631 | 18.473 |
| ResShift | 28.84 | 0.7781 | **0.222** | **8.679** | **17.517** | 27.66 | 0.7461 | 0.184 | 7.965 | **17.422** | 27.23 | **0.7304** | 0.175 | 7.722 | **17.549** |
| InDI | 28.56 | 0.7657 | 0.319 | 9.759 | 19.820 | 27.21 | 0.7125 | 0.199 | 7.984 | 18.861 | 26.40 | 0.6801 | 0.195 | 7.643 | 18.893 |
| BBDM | 28.06 | 0.7529 | 0.278 | 9.157 | 19.533 | 27.14 | 0.7282 | 0.252 | 8.550 | 19.435 | 26.83 | 0.7176 | 0.245 | 8.356 | 19.513 |
| DDBM-VE | 28.55 | 0.7683 | 0.237 | 8.791 | 17.830 | 27.11 | 0.7239 | 0.186 | 7.813 | 17.749 | 26.60 | 0.7021 | 0.178 | 7.556 | 18.079 |
| DDBM-VP | 28.75 | 0.7768 | 0.232 | 8.750 | 17.598 | 27.46 | 0.7410 | 0.190 | 7.966 | 17.584 | 26.98 | 0.7217 | 0.176 | 7.663 | 17.581 |
| I²SB | 28.67 | 0.7708 | 0.245 | 8.960 | 17.938 | 27.46 | 0.7333 | 0.184 | 7.873 | 17.786 | 27.03 | 0.7173 | 0.174 | 7.647 | 17.789 |
| GOUB | 28.52 | 0.7680 | 0.270 | 9.197 | 18.688 | 27.24 | 0.7347 | 0.224 | 8.246 | 18.819 | 26.40 | 0.7136 | 0.213 | 7.922 | 19.076 |
| UniDB | 28.91 | 0.7789 | 0.247 | 8.981 | 17.602 | 27.59 | 0.7387 | **0.181** | 7.883 | 17.589 | 26.28 | 0.7023 | 0.177 | **7.514** | 18.082 |

*Table 6.* Low-light image enhancement on LOL dataset using ancestral sampling with Network forward evaluations (NFE) = 5, 35, and 100. The best values are **bolded** and second to best are underscored.

| Model | NFE = 5 | | | | NFE = 35 | | | | NFE = 100 | | | |
|---|---|---|---|---|---|---|---|---|---|---|---|---|
| | PSNR | SSIM | LPIPS | NIQE | PSNR | SSIM | LPIPS | NIQE | PSNR | SSIM | LPIPS | NIQE |
| DM-VP | 23.01 | 0.6855 | **0.185** | **5.517** | 23.01 | 0.6846 | **0.186** | 5.506 | 23.01 | 0.6842 | **0.186** | 5.514 |
| FM | 22.78 | 0.6848 | 0.202 | 5.677 | 22.57 | 0.6781 | 0.209 | 5.646 | 22.47 | 0.6756 | 0.215 | 5.668 |
| IR-SDE | **23.43** | 0.6960 | 0.208 | 5.745 | 23.07 | 0.6785 | 0.237 | 5.549 | 23.00 | 0.6736 | 0.248 | 5.536 |
| ResShift | 23.13 | 0.6926 | 0.194 | 5.756 | 23.15 | **0.6928** | 0.195 | 5.764 | **23.15** | **0.6928** | 0.195 | 5.753 |
| InDI | 21.26 | 0.6409 | 0.278 | 5.530 | 21.48 | 0.6327 | 0.323 | 5.439 | 21.35 | 0.6250 | 0.334 | 5.487 |
| BBDM | 22.84 | 0.6872 | 0.190 | 5.794 | 22.84 | 0.6869 | 0.191 | 5.788 | 22.84 | 0.6866 | 0.191 | 5.787 |
| DDBM-VE | **23.43** | 0.6933 | 0.201 | 5.555 | **23.18** | 0.6795 | 0.224 | **5.403** | 23.10 | 0.6752 | 0.234 | **5.383** |
| DDBM-VP | 23.16 | 0.6815 | 0.201 | 5.550 | 22.55 | 0.6587 | 0.231 | 5.443 | 22.38 | 0.6520 | 0.243 | 5.447 |
| I²SB | 22.01 | 0.6617 | 0.238 | 5.834 | 21.86 | 0.6552 | 0.245 | 5.769 | 21.86 | 0.6547 | 0.248 | 5.751 |
| GOUB | 23.11 | 0.6919 | 0.207 | 5.599 | 22.57 | 0.6690 | 0.233 | 5.513 | 22.38 | 0.6600 | 0.243 | 5.531 |
| UniDB | 23.36 | **0.6989** | 0.197 | 5.625 | 22.80 | 0.6793 | 0.223 | 5.609 | 22.56 | 0.6715 | 0.235 | 5.592 |

*Table 7.* Image colorization on ImageNet dataset using ancestral sampling with Network forward evaluations (NFE) = 5, 35, and 100. The best values are **bolded** and second to best are underscored. For this task, we also provide FID metric evaluated on 50k samples.

| Model | NFE = 5 | | | | | NFE = 35 | | | | | NFE = 100 | | | | |
|---|---|---|---|---|---|---|---|---|---|---|---|---|---|---|---|
| | PSNR | SSIM | LPIPS | NIQE | FID | PSNR | SSIM | LPIPS | NIQE | FID | PSNR | SSIM | LPIPS | NIQE | FID |
| DM-VP | 22.09 | 0.6969 | 0.186 | 4.772 | 3.859 | 21.73 | 0.6852 | 0.191 | 4.714 | 3.994 | 21.57 | 0.6797 | 0.194 | 4.690 | 4.085 |
| FM | 22.32 | 0.7033 | 0.183 | 4.795 | 3.452 | 21.64 | 0.6850 | 0.191 | 4.762 | 3.500 | 21.31 | 0.6750 | 0.196 | 4.740 | 3.575 |
| IR-SDE | 22.35 | 0.7035 | 0.186 | 4.803 | 3.597 | 22.02 | 0.6905 | 0.187 | 4.730 | 3.751 | 21.82 | 0.6823 | 0.190 | 4.693 | 3.847 |
| ResShift | 22.39 | 0.7058 | 0.183 | 4.822 | 3.473 | 21.82 | 0.6894 | 0.189 | 4.808 | 3.481 | 21.55 | 0.6809 | 0.194 | 4.801 | 3.487 |
| InDI | 22.39 | 0.7044 | 0.192 | 4.811 | 4.449 | 22.28 | 0.6992 | 0.191 | 4.747 | 4.416 | 22.06 | 0.6914 | 0.190 | 4.715 | 4.444 |
| BBDM | 22.44 | 0.7063 | **0.179** | 4.823 | **3.306** | 22.17 | 0.6977 | 0.182 | 4.789 | 3.323 | 21.98 | 0.6916 | 0.185 | 4.780 | **3.324** |
| DDBM-VE | **22.57** | **0.7064** | 0.180 | 4.729 | 3.842 | 22.36 | 0.7003 | **0.181** | 4.737 | 3.712 | 22.17 | 0.6940 | **0.183** | 4.692 | 3.732 |
| DDBM-VP | 22.50 | 0.7052 | 0.182 | **4.671** | 3.487 | 22.23 | 0.6969 | 0.184 | **4.664** | 3.536 | 22.02 | 0.6897 | 0.187 | 4.631 | 3.543 |
| I²SB | 22.52 | 0.7058 | 0.183 | 4.838 | 4.402 | **22.39** | **0.7009** | 0.182 | 4.846 | 4.259 | **22.28** | **0.6972** | **0.183** | 4.830 | 4.208 |
| GOUB | 22.17 | 0.6993 | 0.188 | 4.770 | 3.629 | 21.59 | 0.6808 | 0.193 | 4.691 | 3.799 | 18.54 | 0.5938 | 0.270 | **4.598** | 4.256 |
| UniDB | 22.20 | 0.6999 | 0.186 | 4.787 | 3.637 | 21.69 | 0.6835 | 0.190 | 4.727 | 3.806 | 19.32 | 0.6153 | 0.248 | 4.654 | 4.130 |

*Table 8.* Image deraining on Rain1400 dataset using ancestral sampling with Network forward evaluations (NFE) = 5, 35, and 100. The best values are **bolded** and second to best are underscored.

| Model | NFE = 5 | | | | NFE = 35 | | | | NFE = 100 | | | |
|---|---|---|---|---|---|---|---|---|---|---|---|---|
| | PSNR | SSIM | LPIPS | NIQE | PSNR | SSIM | LPIPS | NIQE | PSNR | SSIM | LPIPS | NIQE |
| DM-VP | 29.92 | 0.8703 | 0.052 | 4.356 | 29.02 | 0.8528 | 0.055 | 4.243 | 28.56 | 0.8450 | 0.058 | 4.239 |
| FM | 29.98 | 0.8702 | 0.050 | 4.290 | 28.89 | 0.8444 | 0.057 | 4.154 | 28.46 | 0.8352 | 0.062 | 4.153 |
| IR-SDE | 29.54 | 0.8627 | 0.048 | 4.217 | 28.15 | 0.8269 | 0.061 | 4.104 | 27.52 | 0.8115 | 0.070 | 4.112 |
| ResShift | 29.69 | 0.8670 | 0.050 | 4.283 | 29.17 | 0.8562 | 0.052 | 4.217 | 28.99 | 0.8527 | 0.052 | 4.197 |
| InDI | 29.93 | 0.8633 | 0.050 | **4.194** | 28.17 | 0.8159 | 0.069 | **4.081** | 27.60 | 0.7995 | 0.079 | **4.087** |
| BBDM | **30.37** | **0.8784** | 0.048 | 4.366 | **30.10** | **0.8731** | **0.049** | 4.349 | **30.04** | **0.8723** | **0.049** | 4.348 |
| DDBM-VE | 29.55 | 0.8595 | 0.050 | 4.293 | 28.65 | 0.8372 | 0.054 | 4.133 | 28.37 | 0.8276 | 0.058 | 4.105 |
| DDBM-VP | 29.82 | 0.8632 | 0.049 | 4.298 | 28.94 | 0.8422 | 0.052 | 4.162 | 28.60 | 0.8315 | 0.056 | 4.109 |
| I²SB | 30.07 | 0.8715 | **0.047** | 4.264 | 29.29 | 0.8542 | 0.050 | 4.165 | 29.14 | 0.8508 | 0.051 | 4.158 |
| GOUB | 29.79 | 0.8673 | 0.050 | 4.277 | 28.38 | 0.8313 | 0.061 | 4.102 | 27.73 | 0.8140 | 0.072 | 4.097 |
| UniDB | 29.38 | 0.8606 | 0.049 | 4.239 | 28.02 | 0.8237 | 0.060 | 4.092 | 27.29 | 0.8031 | 0.073 | 4.116 |

*Table 9.* Image super resolution on DIV2K dataset using ancestral sampling with Network forward evaluations (NFE) = 5, 35, and 100. The best values are **bolded** and second to best are underscored.

| Model | NFE = 5 | | | | NFE = 35 | | | | NFE = 100 | | | |
|---|---|---|---|---|---|---|---|---|---|---|---|---|
| | PSNR | SSIM | LPIPS | NIQE | PSNR | SSIM | LPIPS | NIQE | PSNR | SSIM | LPIPS | NIQE |
| DDPM | 24.22 | 0.6667 | 0.377 | 5.240 | 24.12 | 0.6626 | 0.381 | 5.134 | 24.09 | 0.6608 | 0.383 | 5.104 |
| FM | 24.08 | **0.6767** | 0.386 | 5.223 | 23.80 | 0.6636 | 0.399 | 4.907 | 23.67 | 0.6581 | 0.405 | 4.808 |
| IR-SDE | 23.52 | 0.6464 | 0.398 | 5.044 | 23.01 | 0.6222 | 0.430 | 4.714 | 22.80 | 0.6121 | 0.444 | 4.663 |
| ResShift | **24.42** | 0.6731 | **0.376** | 5.169 | **24.36** | **0.6710** | 0.378 | 5.107 | **24.34** | **0.6703** | **0.378** | 5.088 |
| InDI | 23.53 | 0.6491 | 0.404 | 4.983 | 22.50 | 0.6107 | 0.444 | 4.657 | 22.14 | 0.5983 | 0.456 | **4.597** |
| BBDM | 24.25 | 0.6651 | 0.382 | 5.077 | 24.20 | 0.6636 | 0.384 | 5.040 | 24.18 | 0.6626 | 0.385 | 5.017 |
| DDBM-VE | 24.03 | 0.6673 | 0.378 | 5.073 | 23.95 | 0.6434 | 0.421 | 5.097 | 23.86 | 0.6317 | 0.443 | 5.146 |
| DDBM-VP | 24.38 | 0.6690 | 0.394 | 5.132 | 24.07 | 0.6407 | 0.419 | 5.030 | 23.85 | 0.6229 | 0.446 | 5.085 |
| I$^2$SB | 22.97 | 0.6319 | 0.405 | **4.778** | 22.71 | 0.6190 | 0.418 | **4.653** | 22.65 | 0.6162 | 0.421 | 4.634 |
| GOUB | 23.68 | 0.6573 | 0.388 | 5.121 | 23.30 | 0.6367 | 0.412 | 4.738 | 23.10 | 0.6286 | 0.421 | 4.664 |
| UniDB | 23.71 | 0.6558 | 0.393 | 5.179 | 23.24 | 0.6327 | 0.417 | 4.727 | 23.02 | 0.6243 | 0.427 | 4.655 |

*Table 10.* Aggregated results of all five image enhancement tasks from Table 2. We used three types of aggregation which have different sensitivity to outliers. The aggregated results shows that ResShift is the best method overall and diffusion model is second to best.

| Model | Z-score | | | Min Max Norm | | | Average Rank | | |
|---|---|---|---|---|---|---|---|---|---|
| | PSNR | SSIM | LPIPS | PSNR | SSIM | LPIPS | PSNR | SSIM | LPIPS |
| DM-VP | 0.61 | 0.61 | 0.51 | 0.69 | 0.72 | 0.76 | 8.0 | 7.8 | **8.4** |
| FM | -0.32 | 0.08 | 0.12 | 0.40 | 0.56 | 0.65 | 4.4 | 6.0 | 6.6 |
| IR-SDE | -0.51 | -0.61 | -0.30 | 0.36 | 0.36 | 0.54 | 4.2 | 4.0 | 4.6 |
| ResShift | **0.70** | **0.90** | **0.67** | **0.72** | **0.81** | **0.82** | **9.0** | **9.4** | **8.4** |
| InDI | -0.37 | -0.74 | -0.94 | 0.42 | 0.36 | 0.36 | 5.6 | 4.4 | 3.2 |
| BBDM | 0.13 | 0.34 | -0.15 | 0.54 | 0.64 | 0.58 | 5.8 | 6.8 | 6.6 |
| DDBM-VE | 0.35 | 0.11 | 0.38 | 0.64 | 0.59 | 0.75 | 7.0 | 6.6 | 6.4 |
| DDBM-VP | 0.42 | 0.14 | 0.25 | 0.65 | 0.60 | 0.70 | 6.6 | 6.2 | 5.8 |
| I$^2$SB | -0.06 | -0.02 | 0.40 | 0.50 | 0.55 | 0.76 | 6.6 | 6.0 | 6.4 |
| GOUB | -0.62 | -0.49 | -0.76 | 0.31 | 0.39 | 0.39 | 4.2 | 4.2 | 3.2 |
| UniDB | -0.34 | -0.33 | -0.17 | 0.40 | 0.43 | 0.57 | 4.6 | 4.6 | 6.4 |

*Table 11.* Image super resolution on FFHQ dataset with DIT backbone using ancestral sampling with Network forward evaluations (NFE) = 5, 35, and 100. The best values are **bolded** and second to best are underscored.

| Model | NFE = 5 | | | | | NFE = 35 | | | | | NFE = 100 | | | | |
|---|---|---|---|---|---|---|---|---|---|---|---|---|---|---|---|
| | PSNR | SSIM | LPIPS | NIQE | FID | PSNR | SSIM | LPIPS | NIQE | FID | PSNR | SSIM | LPIPS | NIQE | FID |
| DM-VP | 27.25 | 0.7125 | 0.251 | 4.817 | 22.375 | 26.24 | 0.6761 | 0.265 | 4.379 | 23.074 | 25.80 | 0.6617 | 0.281 | 4.284 | 23.263 |
| FM | 27.27 | 0.7159 | 0.267 | 5.167 | 22.858 | 26.35 | 0.6790 | 0.272 | 4.151 | 23.477 | 25.90 | 0.6540 | 0.289 | 3.781 | 23.810 |
| IR-SDE | 27.94 | **0.7403** | 0.257 | 5.929 | 21.547 | 26.80 | **0.7007** | 0.246 | 4.362 | 22.250 | 26.24 | **0.6782** | 0.262 | 3.964 | 22.423 |
| ResShift | **27.98** | 0.7366 | **0.215** | **4.679** | **20.494** | **26.87** | 0.6941 | **0.218** | **3.660** | **21.197** | 26.45 | 0.6742 | **0.235** | **3.433** | **21.549** |
| InDI | 27.59 | 0.7301 | 0.327 | 7.201 | 23.374 | 26.79 | 0.6946 | 0.289 | 5.123 | 23.873 | 26.34 | 0.6746 | 0.292 | 4.567 | 24.251 |
| BBDM | 27.11 | 0.7102 | 0.272 | 5.036 | 23.262 | 26.46 | 0.6878 | 0.275 | 4.495 | 23.696 | 26.16 | 0.6752 | 0.285 | 4.264 | 24.160 |
| DDBM-VE | 27.33 | 0.7181 | 0.294 | 6.055 | 23.144 | 26.65 | 0.6896 | 0.276 | 4.831 | 23.731 | 26.40 | 0.6731 | 0.280 | 4.423 | 23.801 |
| DDBM-VP | 27.20 | 0.7144 | 0.291 | 5.753 | 23.359 | 26.63 | 0.6893 | 0.283 | 4.779 | 24.160 | 26.44 | 0.6771 | 0.287 | 4.421 | 24.248 |
| I$^2$SB | 27.42 | 0.7213 | 0.293 | 6.084 | 23.167 | 26.73 | 0.6904 | 0.290 | 4.727 | 23.728 | **26.49** | 0.6738 | 0.303 | 4.346 | 24.069 |
| GOUB | 27.71 | 0.7301 | 0.255 | 5.418 | 21.812 | 26.63 | 0.6869 | 0.260 | 3.990 | 22.707 | 25.48 | 0.6503 | 0.297 | 3.491 | 23.261 |
| UniDB | 27.75 | 0.7321 | 0.260 | 5.689 | 22.059 | 26.81 | 0.6949 | 0.260 | 4.263 | 22.800 | 25.87 | 0.6642 | 0.282 | 3.638 | 23.154 |

*Table 12.* Image super resolution on FFHQ dataset with original hyperparameter values with Network forward evaluations (NFE) = 5, 35, and 100. The best values are **bolded** and second to best are underscored. We used the same samplers as the original works.

| Model | Sampler | NFE = 5 | | | | | NFE = 35 | | | | | NFE = 100 | | | | |
|---|---|---|---|---|---|---|---|---|---|---|---|---|---|---|---|---|
| | | PSNR | SSIM | LPIPS | NIQE | FID | PSNR | SSIM | LPIPS | NIQE | FID | PSNR | SSIM | LPIPS | NIQE | FID |
| DM-VP | Ancestral | 23.22 | 0.645 | 0.312 | 7.65 | 19.06 | 26.13 | 0.712 | 0.248 | 5.99 | 18.31 | 26.02 | 0.701 | 0.218 | 4.95 | 18.24 |
| FM | Euler ODE | 28.47 | 0.769 | 0.233 | 6.09 | 17.35 | 27.01 | 0.716 | **0.169** | 3.96 | 17.40 | 26.70 | 0.697 | **0.168** | **3.63** | 17.57 |
| IR-SDE | Euler ODE | 22.98 | 0.290 | 0.710 | 6.20 | 23.46 | 24.87 | 0.533 | 0.347 | 3.98 | 19.85 | 24.82 | 0.595 | 0.264 | 3.82 | 19.26 |
| | Euler SDE | 9.41 | 0.006 | 1.510 | 15.16 | 34.71 | 17.79 | 0.147 | 0.943 | 4.64 | 23.50 | 22.89 | 0.401 | 0.461 | 3.70 | 20.17 |
| ResShift | Ancestral | 28.84 | 0.778 | **0.222** | 8.68 | 17.52 | 27.66 | 0.746 | 0.184 | 7.97 | 17.42 | 27.23 | 0.730 | 0.175 | 7.72 | 17.55 |
| InDI | Euler ODE | 28.69 | 0.774 | 0.257 | 6.98 | 17.94 | 27.16 | 0.714 | 0.170 | 4.04 | 17.84 | 26.59 | 0.696 | 0.174 | 3.78 | 17.97 |
| BBDM | Ancestral | 24.60 | 0.589 | 0.277 | 5.68 | 19.90 | 27.85 | 0.757 | 0.229 | 5.82 | 17.57 | 27.68 | 0.749 | 0.213 | 5.34 | **17.54** |
| DDBM-VE | Langevin-Heun | 29.29 | 0.790 | 0.290 | 7.88 | 18.41 | 21.18 | 0.229 | 0.813 | 4.34 | 25.42 | 26.62 | 0.671 | 0.240 | 4.00 | 19.14 |
| | Heun | 9.55 | 0.057 | 0.704 | **4.50** | 31.19 | 26.61 | 0.752 | 0.335 | 7.74 | 21.42 | 28.41 | 0.780 | 0.315 | 7.98 | 20.42 |
| DDBM-VP | Langevin-Heun | **29.46** | **0.795** | 0.279 | 7.65 | 17.97 | 18.42 | 0.210 | 0.935 | 5.62 | 27.42 | 26.62 | 0.634 | 0.300 | 4.43 | 19.75 |
| | Heun | 12.23 | 0.434 | 0.631 | 9.86 | 32.03 | **28.05** | **0.758** | 0.321 | 7.44 | 21.37 | **29.05** | **0.786** | 0.304 | 7.82 | 19.85 |
| I$^2$SB | Ancestral | 13.04 | 0.052 | 0.887 | 8.53 | 32.40 | 19.25 | 0.516 | 0.325 | 5.26 | 20.50 | 20.10 | 0.553 | 0.316 | 6.15 | 19.76 |
| GOUB | Mean ODE | 17.92 | 0.170 | 0.729 | 10.70 | 27.27 | 23.38 | 0.674 | 0.392 | 6.90 | 23.66 | 20.83 | 0.662 | 0.439 | 8.06 | 24.73 |
| | Euler ODE | 21.98 | 0.393 | 0.584 | 8.96 | 24.75 | 24.16 | 0.693 | 0.404 | 7.50 | 22.87 | 23.13 | 0.690 | 0.416 | 7.88 | 23.19 |
| UniDB | Mean ODE | 17.81 | 0.168 | 0.736 | 10.28 | 27.71 | 24.20 | 0.684 | 0.375 | 7.49 | 23.54 | 21.82 | 0.672 | 0.410 | 8.02 | 24.18 |
| | Euler ODE | 21.83 | 0.385 | 0.591 | 8.93 | 24.82 | 25.21 | 0.713 | 0.378 | 7.76 | 22.68 | 24.61 | 0.714 | 0.378 | 8.13 | 22.69 |

*Table 13.* Different samplers used in image super-resolution for diffusion and Ornstein-Uhlenbeck processes. We measure their effectiveness using PSNR / SSIM / LPIPS metrics for Network Forward Evaluations (NFE) = 5, 35, and 100. Best results across samplers are **bolded** and second best are underscored. Ancestral sampling and Euler ODE achieve the best LPIPS results. Mean-reverting ODE yields the highest PSNR and SSIM but low LPIPS, suggesting possible oversmoothing. Our Exponential Integrator (EI-ODE) relies on numerical methods, which perform worse for flow matching and InDI due to their stiff dynamics. Surprisingly, Heun performs worst among the second-order Runge-Kutta methods.

| NFE | Sampler | DM-VP | FM | IR-SDE | ResShift | InDI |
|---|---|---|---|---|---|---|
| 5 | Euler ODE | 25.2 / 0.51 / 0.40 | 28.4 / 0.77 / **0.22** | 26.3 / 0.58 / 0.35 | 28.2 / 0.76 / **0.20** | 28.5 / 0.76 / **0.29** |
| | Euler SDE | 17.5 / 0.18 / 1.00 | 8.9 / 0.01 / 0.84 | 18.7 / 0.13 / 0.82 | 18.5 / 0.15 / 1.07 | 7.6 / 0.00 / 1.49 |
| | Ancestral† | 29.0 / 0.78 / **0.23** | 28.7 / 0.78 / 0.24 | 28.6 / **0.77** / **0.25** | 28.8 / 0.78 / 0.22 | 28.6 / **0.77** / 0.32 |
| | EI-ODE | 14.9 / 0.11 / 0.97 | 9.5 / 0.01 / 0.84 | 14.5 / 0.13 / 0.75 | 18.1 / 0.18 / 0.94 | 3.0 / -0.39 / 0.82 |
| | Mean-ODE†† | 21.4 / 0.29 / 0.79 | 8.9 / 0.01 / 0.84 | 21.7 / 0.23 / 0.60 | 20.0 / 0.19 / 1.00 | 7.6 / 0.00 / 1.49 |
| | Langevin-Heun††† | **29.5 / 0.79** / 0.28 | **29.3 / 0.79** / 0.29 | **29.2 / 0.79** / 0.30 | **29.5 / 0.79** / 0.28 | **28.7 / 0.77** / 0.34 |
| | 2nd Heun | 15.8 / 0.13 / 0.92 | 13.0 / 0.30 / 0.61 | 9.0 / 0.01 / 0.94 | 12.5 / 0.03 / 0.87 | 12.9 / 0.33 / 0.59 |
| | 2nd Midpoint | 20.7 / 0.34 / 0.66 | 28.1 / 0.76 / 0.23 | 25.9 / 0.62 / **0.25** | 24.6 / 0.47 / 0.43 | 28.5 / 0.76 / 0.30 |
| | 2nd Ralston | 20.4 / 0.60 / 0.37 | 27.9 / 0.75 / **0.22** | 22.7 / 0.27 / 0.57 | 22.5 / 0.31 / 0.65 | 28.6 / **0.77** / 0.31 |
| 35 | Euler ODE | 27.0 / 0.71 / **0.16** | 26.7 / 0.71 / **0.17** | 26.4 / 0.68 / 0.19 | 27.2 / 0.73 / **0.18** | 26.8 / 0.69 / **0.19** |
| | Euler SDE | 27.5 / 0.70 / 0.25 | 25.5 / 0.55 / 0.29 | 25.7 / 0.62 / 0.25 | 27.5 / 0.74 / 0.19 | 16.7 / 0.15 / 0.84 |
| | Ancestral | 27.9 / 0.75 / 0.18 | 27.2 / 0.73 / 0.18 | 27.0 / 0.72 / 0.18 | 27.7 / 0.75 / 0.18 | **27.2 / 0.71** / 0.20 |
| | EI-ODE | 27.3 / 0.73 / 0.25 | 13.1 / 0.17 / 0.75 | 25.6 / 0.70 / 0.21 | 27.3 / 0.75 / 0.22 | 11.5 / 0.44 / 0.64 |
| | Mean-ODE | **28.2 / 0.78** / 0.32 | **28.2 / 0.78** / 0.32 | **27.9 / 0.77** / 0.32 | **28.2 / 0.78** / 0.30 | 25.8 / 0.67 / 0.45 |
| | Langevin-Heun | 25.2 / 0.58 / 0.38 | 12.6 / 0.19 / 0.91 | 22.8 / 0.48 / 0.35 | 19.8 / 0.24 / 0.74 | 5.8 / -0.00 / 0.92 |
| | 2nd Heun | 25.6 / 0.63 / 0.28 | 23.2 / 0.58 / 0.27 | 23.8 / 0.57 / 0.26 | 25.1 / 0.44 / 0.62 | 24.8 / 0.65 / 0.23 |
| | 2nd Midpoint | 26.9 / 0.70 / 0.17 | 26.3 / 0.69 / 0.18 | 26.1 / 0.67 / 0.19 | 26.9 / 0.72 / **0.18** | 26.1 / 0.66 / 0.20 |
| | 2nd Ralston | 26.8 / 0.70 / 0.17 | 26.2 / 0.68 / 0.18 | 26.1 / 0.67 / 0.19 | 26.9 / 0.72 / **0.18** | 26.4 / 0.67 / 0.20 |
| 100 | Euler ODE | 26.8 / 0.69 / 0.17 | 26.4 / 0.69 / **0.17** | 26.1 / 0.67 / **0.19** | 27.0 / 0.72 / 0.18 | 26.2 / 0.67 / 0.20 |
| | Euler SDE | 27.2 / 0.73 / 0.18 | 25.9 / 0.65 / 0.20 | 25.6 / 0.65 / 0.21 | 27.2 / 0.73 / 0.18 | 21.6 / 0.50 / 0.39 |
| | Ancestral | 27.2 / 0.72 / **0.16** | 26.5 / 0.70 / **0.17** | 26.2 / 0.69 / **0.19** | 27.2 / 0.73 / **0.17** | 26.4 / 0.68 / **0.19** |
| | EI-ODE | 26.9 / 0.72 / 0.18 | 17.0 / 0.21 / 0.69 | 25.8 / 0.68 / **0.19** | 27.0 / 0.72 / **0.17** | 13.1 / 0.48 / 0.56 |
| | Mean-ODE | **28.2 / 0.78** / 0.32 | **28.2 / 0.78** / 0.32 | **27.6 / 0.77** / 0.33 | **28.1 / 0.78** / 0.30 | **27.3 / 0.75** / 0.35 |
| | Langevin-Heun | 26.4 / 0.67 / 0.19 | 13.6 / 0.17 / 0.70 | 25.1 / 0.63 / 0.23 | 27.1 / 0.72 / 0.25 | 8.6 / 0.04 / 0.99 |
| | 2nd Heun | 26.3 / 0.67 / 0.19 | 25.3 / 0.65 / 0.21 | 25.3 / 0.65 / 0.21 | 27.3 / 0.73 / 0.24 | 25.4 / 0.65 / 0.21 |
| | 2nd Midpoint | 26.6 / 0.68 / 0.18 | 26.2 / 0.67 / 0.18 | 25.9 / 0.67 / 0.20 | 26.8 / 0.71 / 0.18 | 25.8 / 0.65 / 0.20 |
| | 2nd Ralston | 26.6 / 0.68 / 0.18 | 26.1 / 0.67 / 0.18 | 25.9 / 0.67 / 0.20 | 26.8 / 0.71 / **0.17** | 25.8 / 0.65 / 0.20 |

† Ancestral sampling of the Markov model that discretizes forward SDE (see (Ho et al., 2020)).
†† See (Yue et al., 2023a).
††† See (Zhou et al., 2023b).

*Table 14.* Different samplers used in image super-resolution for diffusion bridges. We measure their effectiveness using PSNR / SSIM / LPIPS metrics for Network Forward Evaluations (NFE) = 5, 35, and 100. Best results across samplers are **bolded** and second best are underscored. Ancestral sampling remains the best choice globally. Note that no deterministic sampler achieves an LPIPS below 0.3, even with 100 NFEs.

| NFE | Sampler | BBDM | DDBM-VE | DDBM-VP | I$^2$SB | GOUB | UniDB |
|---|---|---|---|---|---|---|---|
| 5 | Euler ODE | 28.5 / **0.77** / 0.36 | 28.6 / 0.78 / 0.31 | 28.7 / **0.79** / 0.30 | 28.8 / 0.78 / 0.34 | 28.7 / 0.77 / 0.34 | 29.2 / **0.79** / 0.32 |
| | Euler SDE | 27.4 / 0.72 / **0.25** | 14.0 / 0.05 / 1.22 | 11.6 / 0.03 / 1.35 | 23.2 / 0.31 / 0.74 | 17.6 / 0.10 / 0.89 | 13.2 / 0.09 / 0.86 |
| | Ancestral† | 28.1 / 0.75 / 0.28 | 28.5 / 0.77 / **0.24** | 28.8 / 0.78 / **0.23** | 28.7 / 0.77 / 0.25 | 28.5 / 0.77 / **0.27** | 28.9 / 0.78 / **0.25** |
| | EI-ODE | 3.4 / -0.37 / 0.86 | 2.6 / -0.43 / 0.83 | 2.6 / -0.43 / 0.83 | 2.9 / -0.41 / 0.84 | 12.8 / 0.48 / 0.60 | 12.9 / 0.49 / 0.59 |
| | Mean-ODE†† | 28.5 / **0.77** / 0.34 | 25.8 / 0.74 / 0.34 | 27.1 / 0.77 / 0.32 | 29.0 / **0.79** / 0.31 | 28.3 / 0.77 / 0.32 | 28.9 / 0.78 / 0.31 |
| | Langevin-Heun††† | **28.7** / **0.77** / 0.35 | 29.2 / 0.79 / 0.30 | 29.4 / 0.79 / 0.28 | 29.2 / 0.79 / 0.30 | 29.0 / 0.78 / 0.31 | 29.4 / **0.79** / 0.29 |
| | 2nd Heun | 11.6 / 0.22 / 0.73 | 16.9 / 0.19 / 0.72 | 12.9 / 0.41 / 0.59 | 9.9 / 0.08 / 0.73 | 9.9 / 0.15 / 0.74 | 11.4 / 0.24 / 0.69 |
| | 2nd Midpoint | 28.6 / **0.77** / 0.35 | 27.1 / 0.75 / 0.33 | 21.8 / 0.74 / 0.33 | 28.8 / 0.78 / 0.35 | 27.0 / 0.73 / 0.38 | 27.4 / 0.75 / 0.34 |
| | 2nd Ralston | 28.4 / 0.76 / 0.36 | 27.9 / 0.75 / 0.33 | 21.5 / 0.72 / 0.36 | 28.4 / 0.77 / 0.38 | 26.9 / 0.72 / 0.45 | 27.3 / 0.73 / 0.42 |
| 35 | Euler ODE | **28.5** / **0.77** / 0.35 | **27.8** / **0.77** / 0.32 | **28.5** / 0.78 / 0.31 | **28.9** / 0.78 / 0.31 | **28.6** / 0.77 / 0.32 | **29.3** / **0.79** / 0.31 |
| | Euler SDE | 27.2 / 0.73 / 0.26 | 24.2 / 0.39 / 0.43 | 23.0 / 0.31 / 0.53 | 27.1 / 0.70 / **0.18** | 26.4 / 0.67 / **0.20** | 25.5 / 0.56 / 0.31 |
| | Ancestral | 27.1 / 0.73 / 0.25 | 27.1 / 0.72 / **0.19** | 27.5 / 0.74 / **0.19** | 27.5 / 0.73 / **0.18** | 27.2 / 0.73 / 0.22 | 27.6 / 0.74 / **0.18** |
| | EI-ODE | 10.6 / 0.24 / 0.71 | 12.5 / 0.35 / 0.70 | 13.1 / 0.36 / 0.68 | 4.3 / -0.25 / 0.81 | 15.5 / 0.42 / 0.68 | 11.9 / 0.26 / 0.67 |
| | Mean-ODE | 28.2 / 0.76 / 0.34 | 24.3 / 0.73 / 0.36 | 26.3 / 0.76 / 0.34 | 25.5 / 0.75 / 0.34 | 27.8 / 0.76 / 0.33 | 28.9 / 0.78 / 0.32 |
| | Langevin-Heun | 14.7 / 0.08 / 0.92 | 21.7 / 0.27 / 0.46 | 19.5 / 0.19 / 0.56 | 19.6 / 0.17 / 0.93 | 23.4 / 0.45 / 0.31 | 23.6 / 0.46 / 0.36 |
| | 2nd Heun | 25.7 / 0.70 / 0.39 | 27.6 / 0.76 / 0.32 | 27.9 / 0.76 / 0.31 | 27.3 / 0.75 / 0.32 | 25.9 / 0.71 / 0.36 | 27.5 / 0.75 / 0.32 |
| | 2nd Midpoint | **28.5** / 0.76 / 0.34 | 27.5 / **0.77** / 0.32 | 28.4 / **0.78** / 0.31 | 28.5 / **0.78** / 0.31 | 28.5 / **0.77** / 0.32 | 29.2 / **0.79** / 0.31 |
| | 2nd Ralston | 28.4 / 0.76 / 0.35 | 27.6 / **0.77** / 0.32 | 28.4 / **0.78** / 0.31 | 28.6 / **0.78** / 0.31 | 28.5 / **0.77** / 0.32 | 29.2 / **0.79** / 0.31 |
| 100 | Euler ODE | **28.4** / 0.76 / 0.34 | **27.5** / **0.77** / 0.32 | **28.4** / **0.78** / 0.31 | **28.2** / 0.78 / 0.31 | **28.5** / 0.77 / 0.32 | 29.2 / **0.79** / 0.31 |
| | Euler SDE | 26.9 / 0.72 / 0.25 | 25.6 / 0.56 / 0.28 | 25.3 / 0.51 / 0.32 | 26.9 / 0.71 / **0.17** | 26.4 / 0.70 / **0.21** | 26.3 / 0.66 / 0.20 |
| | Ancestral | 26.8 / 0.72 / **0.24** | 26.6 / 0.70 / **0.18** | 27.0 / 0.72 / **0.18** | 27.0 / 0.72 / **0.17** | 26.4 / 0.71 / **0.21** | 26.3 / 0.70 / **0.18** |
| | EI-ODE | 21.7 / 0.68 / 0.48 | 14.7 / 0.54 / 0.53 | 16.5 / 0.60 / 0.48 | 12.1 / 0.47 / 0.63 | 16.0 / 0.45 / 0.66 | 13.8 / 0.33 / 0.64 |
| | Mean-ODE | 28.1 / **0.76** / 0.34 | 24.1 / 0.73 / 0.37 | 26.1 / 0.75 / 0.34 | 24.1 / 0.74 / 0.36 | 27.6 / 0.76 / 0.34 | 28.9 / 0.78 / 0.32 |
| | Langevin-Heun | 27.2 / 0.69 / 0.37 | 25.8 / 0.62 / 0.24 | 25.7 / 0.58 / 0.27 | 26.8 / 0.67 / 0.26 | 25.9 / 0.69 / **0.21** | 26.0 / 0.66 / 0.20 |
| | 2nd Heun | 27.7 / 0.75 / 0.36 | 27.4 / **0.77** / 0.33 | 28.3 / **0.78** / 0.31 | 27.8 / 0.77 / 0.32 | 27.9 / 0.76 / 0.33 | 28.9 / 0.78 / 0.31 |
| | 2nd Midpoint | **28.4** / 0.76 / 0.34 | 27.3 / **0.77** / 0.32 | 28.3 / **0.78** / 0.31 | 27.7 / **0.78** / 0.32 | **28.5** / 0.77 / 0.32 | 29.2 / **0.79** / 0.31 |
| | 2nd Ralston | **28.4** / 0.76 / 0.34 | 27.3 / **0.77** / 0.32 | 28.3 / **0.78** / 0.31 | 27.9 / **0.78** / 0.32 | 28.4 / **0.77** / 0.32 | 29.2 / **0.79** / 0.31 |

† Ancestral sampling of the Markov model that discretizes forward SDE (see (Ho et al., 2020)).
†† See (Yue et al., 2023a).
††† See (Zhou et al., 2023b).

*Table 15.* Alignment of each method group (unconditional processes, Ornstein–Uhlenbeck processes, and Diffusion Bridges) with different image enhancement tasks. We first compute Z-scores across all methods, followed by Z-scores across tasks. The results are then averaged within each method group. This procedure measures the relative performance of each group on each task.

| Methods | Facial Super-Resolution | | | Low-light enhancement | | | Colorization | | | Deraining | | | General Super-Resolution | | |
|---|---|---|---|---|---|---|---|---|---|---|---|---|---|---|---|
| | PSNR | SSIM | LPIPS | PSNR | SSIM | LPIPS | PSNR | SSIM | LPIPS | PSNR | SSIM | LPIPS | PSNR | SSIM | LPIPS |
| Unconditional processes | **0.59** | **0.64** | **0.62** | **0.28** | **0.55** | **0.83** | -1.05 | -0.97 | -0.87 | **0.27** | **0.44** | 0.06 | **0.64** | **1.07** | **0.95** |
| Ornstein-Uhlenbeck processes | -0.27 | -0.62 | 0.29 | -0.15 | -0.25 | -0.69 | 0.16 | 0.17 | -0.39 | -0.52 | -0.54 | -0.76 | 0.47 | 0.5 | 0.59 |
| Diffusion bridges | -0.06 | 0.1 | -0.35 | -0.02 | -0.06 | 0.07 | **0.27** | **0.24** | **0.49** | 0.17 | 0.12 | **0.36** | -0.45 | -0.6 | -0.61 |

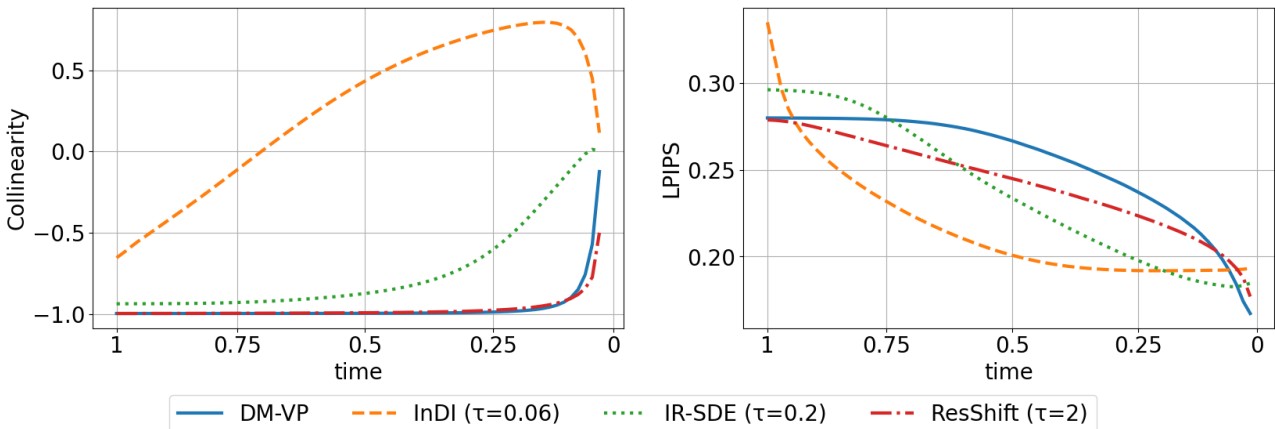

*Figure 4.* Comparison of diffusion and three OU processes: InDI, IR-SDE, and ResShift with progressively higher temperature parameter $\tau$, to study how temperature influences the collinearity of $\mathbf{x}_t, \mathbf{y}, \hat{\mathbf{x}}_{0|t}$, and how this collinearity, in turn, affects the LPIPS metric. Left: collinearity of normalized vectors $\mathbf{x}_t - \mathbf{y}$ and $\hat{\mathbf{x}}_{0|t} - \mathbf{x}_t$. InDI, with the lowest $\tau$, is trained to extrapolate the $\mathbf{x}_t - \mathbf{y}$ vector, especially at the end of the trajectory, where the transition variance was very small during training. Right: LPIPS of predicted $\hat{\mathbf{x}}_{0|t}$ for different $t$. It can be noticed that when InDI has high collinearity, it does not improve over time. Later steps are redundant or even harmful to the quality of the final sample. Experiments were conducted on image super-resolution.

*Table 16.* Different discretization strategies (see Figure 2) for image super-resolution. The results show that model quality does not depend on the chosen method. We report PSNR / SSIM / LPIPS. The best values are **bolded** and second to best are underscored.

| NFE | Method | Linear | Karras† | DDBM†† |
|-----|--------|--------|---------|--------|
| 5 | DM-VP | 29.0 / 0.78 / 0.23 | 28.9 / 0.77 / **0.20** | **29.4** / **0.79** / 0.26 |
| | FM | 28.7 / 0.78 / **0.24** | 28.9 / 0.78 / 0.24 | **29.1** / **0.79** / 0.27 |
| | ResShift | 28.8 / **0.78** / 0.22 | 28.8 / 0.77 / 0.21 | **29.0** / **0.78** / 0.23 |
| | IR-SDE | 28.6 / 0.77 / **0.25** | **28.9** / **0.78** / 0.27 | **28.9** / **0.78** / 0.28 |
| | InDI | 28.6 / 0.77 / 0.32 | **28.7** / **0.77** / 0.33 | 28.5 / 0.77 / **0.31** |
| | BBDM | 28.1 / 0.75 / **0.28** | 28.1 / 0.75 / **0.28** | **28.4** / **0.76** / 0.30 |
| | DDBM-VE | 28.5 / 0.77 / **0.24** | **28.8** / 0.77 / 0.25 | **28.8** / **0.78** / 0.25 |
| | DDBM-VP | 28.8 / 0.78 / **0.23** | 29.0 / 0.78 / 0.24 | **29.1** / **0.79** / 0.25 |
| | I²SB | 28.7 / 0.77 / **0.25** | **28.9** / **0.78** / 0.26 | 28.8 / **0.78** / 0.25 |
| | GOUB | 28.5 / 0.77 / **0.27** | 28.7 / 0.77 / 0.28 | **28.9** / **0.78** / 0.30 |
| | UniDB | 28.9 / 0.78 / **0.25** | 29.1 / 0.78 / 0.26 | **29.3** / **0.79** / 0.27 |
| 35 | DM-VP | **27.9** / **0.75** / 0.18 | 27.3 / 0.71 / **0.17** | 27.8 / 0.73 / **0.17** |
| | FM | 27.2 / 0.73 / 0.18 | 27.1 / 0.71 / **0.17** | **27.7** / **0.74** / **0.17** |
| | IR-SDE | 27.0 / 0.72 / **0.18** | 27.2 / 0.72 / **0.18** | **27.9** / **0.75** / 0.20 |
| | ResShift | 27.7 / **0.75** / **0.18** | 27.5 / 0.73 / **0.17** | **27.8** / **0.74** / **0.17** |
| | InDI | 27.2 / 0.71 / **0.20** | **27.8** / **0.74** / 0.25 | 27.1 / 0.73 / 0.23 |
| | BBDM | **27.1** / **0.73** / 0.25 | 26.9 / 0.72 / 0.24 | **27.1** / 0.72 / 0.25 |
| | DDBM-VE | 27.1 / 0.72 / **0.19** | 27.3 / 0.72 / **0.17** | **27.5** / **0.73** / **0.17** |
| | DDBM-VP | 27.5 / **0.74** / 0.19 | 27.4 / 0.72 / **0.16** | **27.8** / **0.74** / 0.17 |
| | I²SB | 27.5 / 0.73 / **0.18** | **27.7** / 0.73 / **0.18** | 27.6 / **0.74** / **0.18** |
| | GOUB | 27.2 / 0.73 / **0.22** | 27.4 / 0.74 / 0.23 | **27.8** / **0.75** / 0.24 |
| | UniDB | 27.6 / 0.74 / **0.18** | 27.9 / 0.74 / **0.18** | **28.4** / **0.76** / 0.20 |
| 100 | DM-VP | **27.2** / **0.72** / **0.16** | 26.7 / 0.68 / 0.19 | 27.0 / 0.69 / 0.18 |
| | FM | 26.5 / **0.70** / **0.17** | 26.2 / 0.68 / 0.18 | **26.7** / **0.70** / **0.17** |
| | IR-SDE | 26.2 / 0.69 / **0.19** | 26.2 / 0.69 / **0.19** | **26.7** / **0.71** / **0.18** |
| | ResShift | **27.2** / **0.73** / **0.17** | 27.1 / 0.72 / **0.17** | **27.2** / 0.72 / **0.17** |
| | InDI | 26.4 / 0.68 / **0.19** | **26.9** / **0.71** / 0.21 | 26.1 / 0.69 / 0.20 |
| | BBDM | **26.8** / **0.72** / 0.24 | 26.6 / 0.71 / 0.24 | 26.7 / 0.71 / **0.24** |
| | DDBM-VE | 26.6 / **0.70** / 0.18 | 26.7 / 0.69 / **0.17** | **26.8** / **0.70** / **0.17** |
| | DDBM-VP | **27.0** / **0.72** / 0.18 | 26.8 / 0.70 / 0.17 | **27.0** / **0.71** / **0.16** |
| | I²SB | 27.0 / **0.72** / **0.17** | **27.2** / 0.71 / **0.17** | 27.0 / 0.71 / **0.17** |
| | GOUB | 26.4 / 0.71 / **0.21** | **26.7** / **0.72** / 0.22 | 16.0 / 0.41 / 0.53 |
| | UniDB | 26.3 / 0.70 / **0.18** | **27.2** / **0.71** / **0.17** | 13.3 / 0.23 / 0.67 |

† See (Karras et al., 2022) Appendix D.1.
†† See (Zhou et al., 2023b). Formulation is taken from the official Github repository.

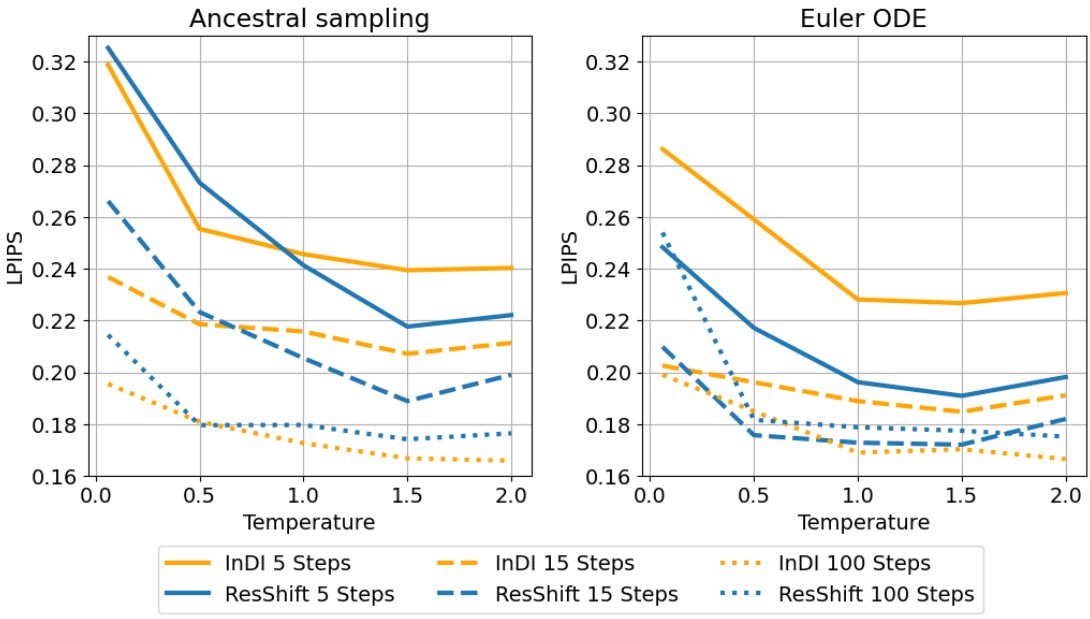

*Figure 5.* Image quality for the InDI and ResShift models across different temperature settings. Image quality generally improves as the temperature increases. With the default temperature values, ResShift produces noticeably higher-quality results than InDI. The results are grouped by the number of diffusion steps used during inference. On the left, we used ancestral sampling, which is the original sampler in ResShift, and on the right, we used Euler ODE, the original sampler from InDI.

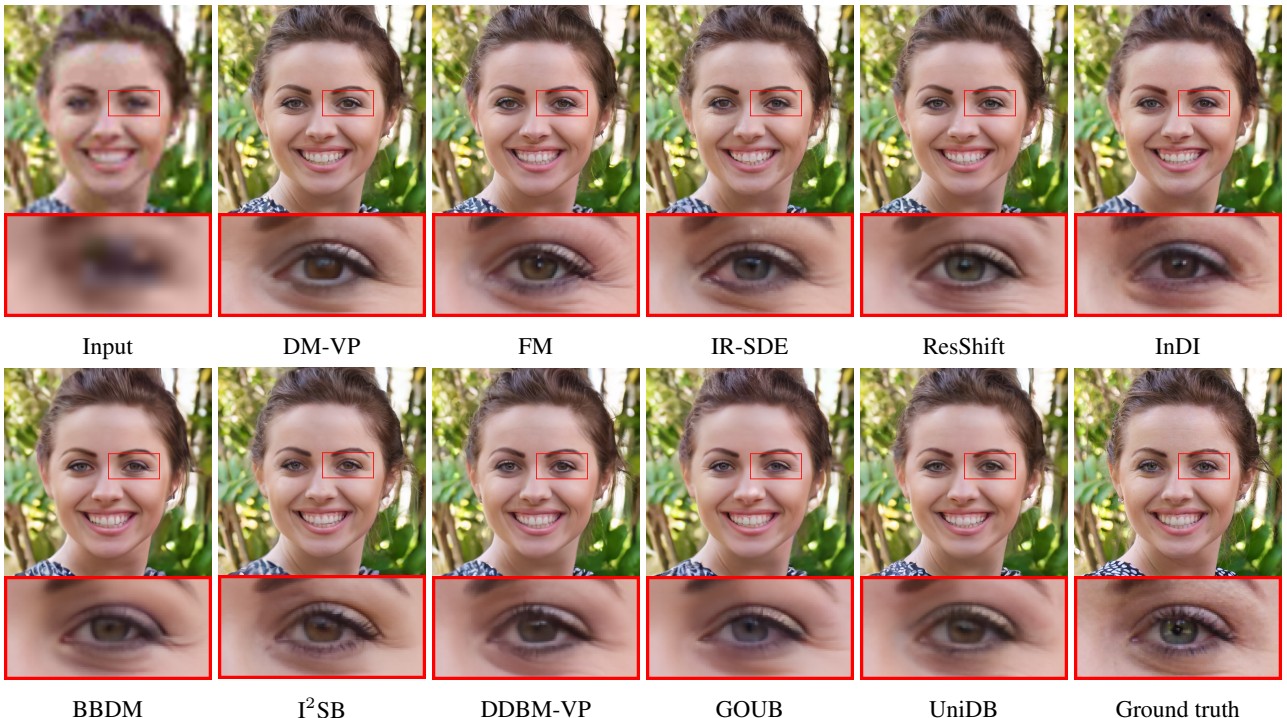

*Figure 6.* Visual comparison of image super resolution results performed on FFHQ dataset. Results were generated using ancestral sampling with 35 steps. All methods achieve similar visual quality, except BBDM and GOUB, which produce slightly blurred outputs.

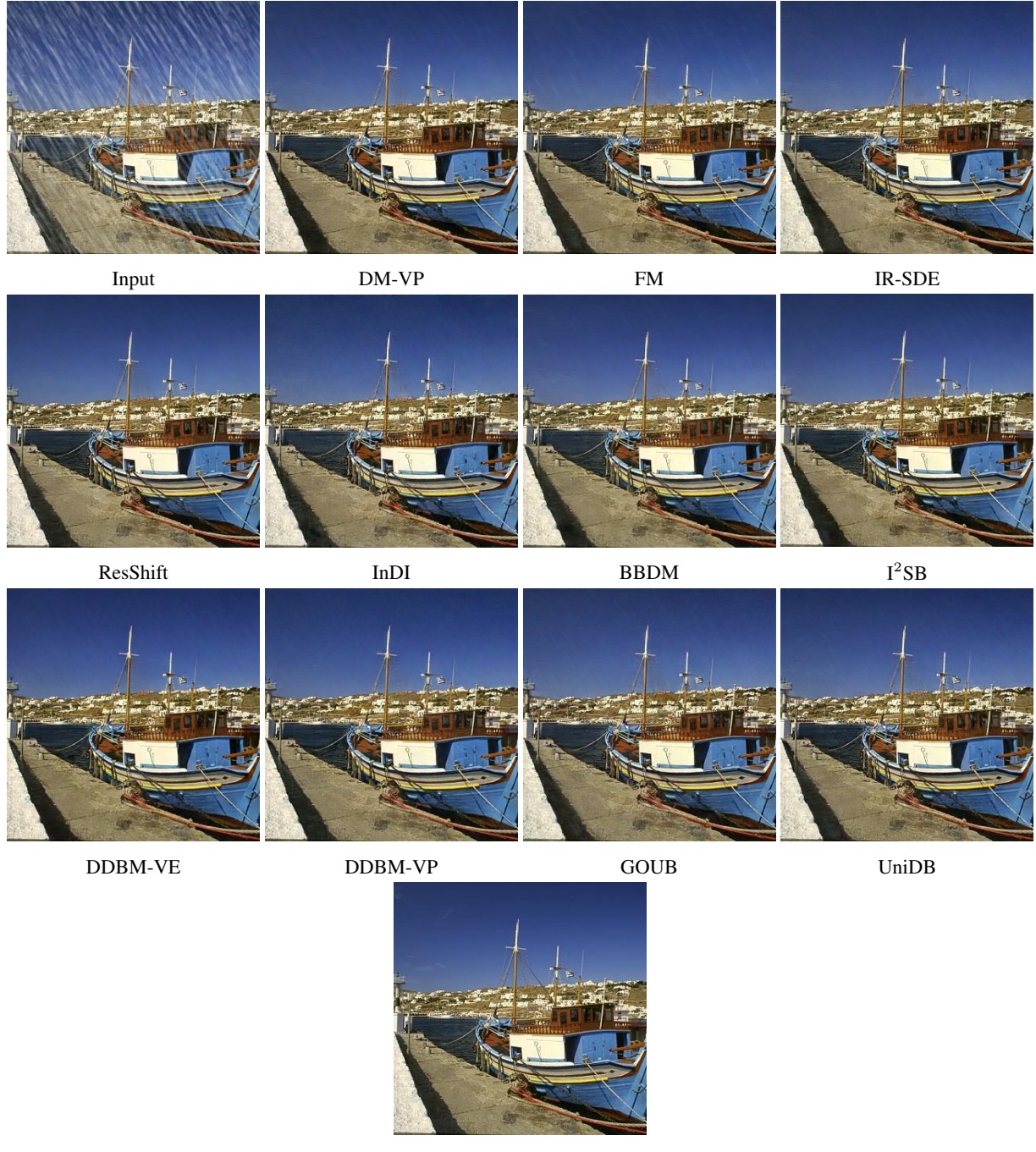

Input      DM-VP      FM      IR-SDE

ResShift      InDI      BBDM      I²SB

DDBM-VE      DDBM-VP      GOUB      UniDB

Ground truth

*Figure 7.* Visual comparison of image deraining on the Rain1400 dataset using ancestral sampling with 35 steps. All outputs appear similar, but diffusion and flow matching tend to leave slightly more visible traces of raindrops.

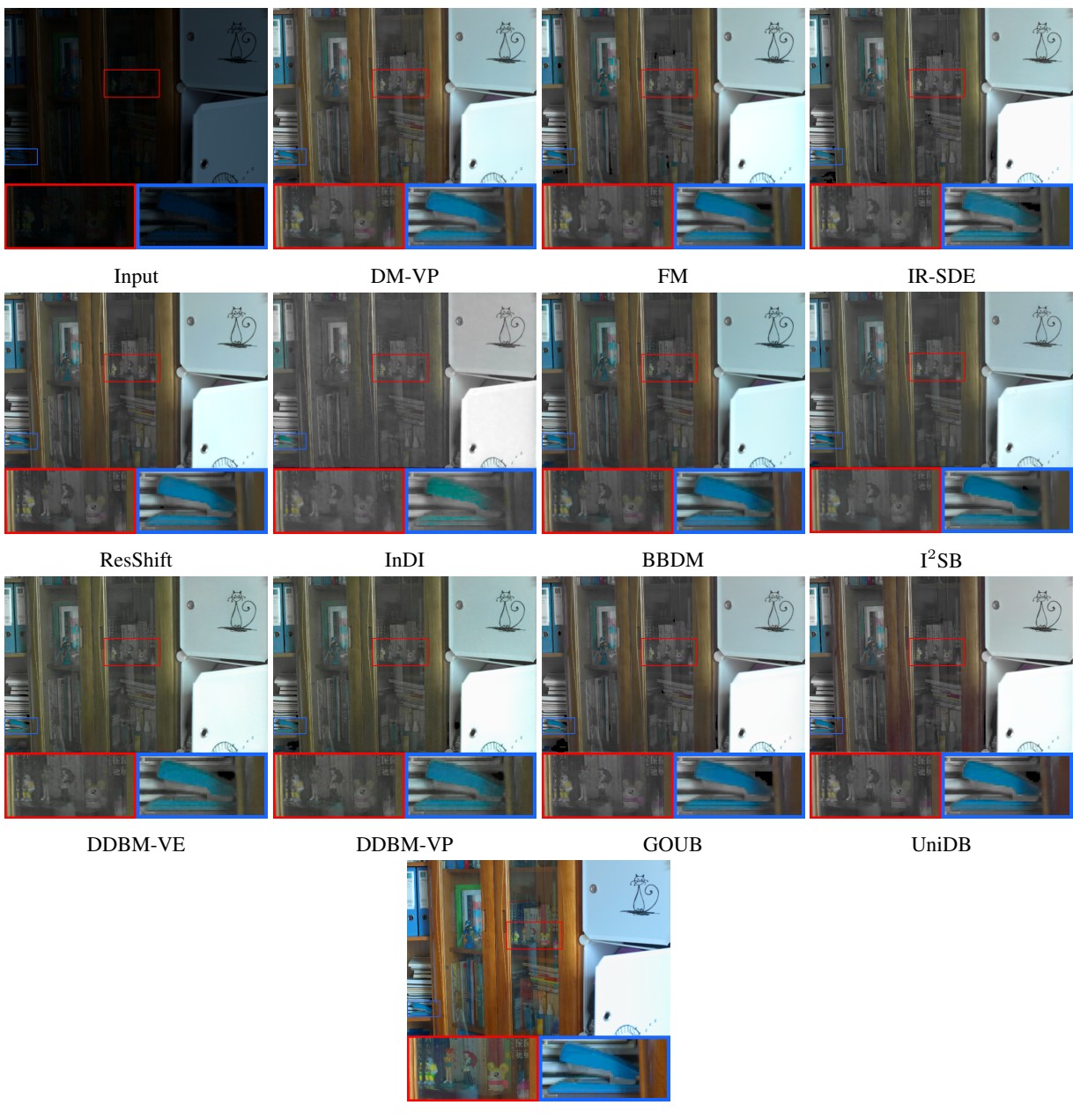

*Figure 8.* Visual comparison of low-light image enhancement on the LOL dataset using ancestral sampling with 35 steps. Most methods produce images of similar quality, while InDI performs noticeably worse. All methods still struggle to fully recover the ground truth.

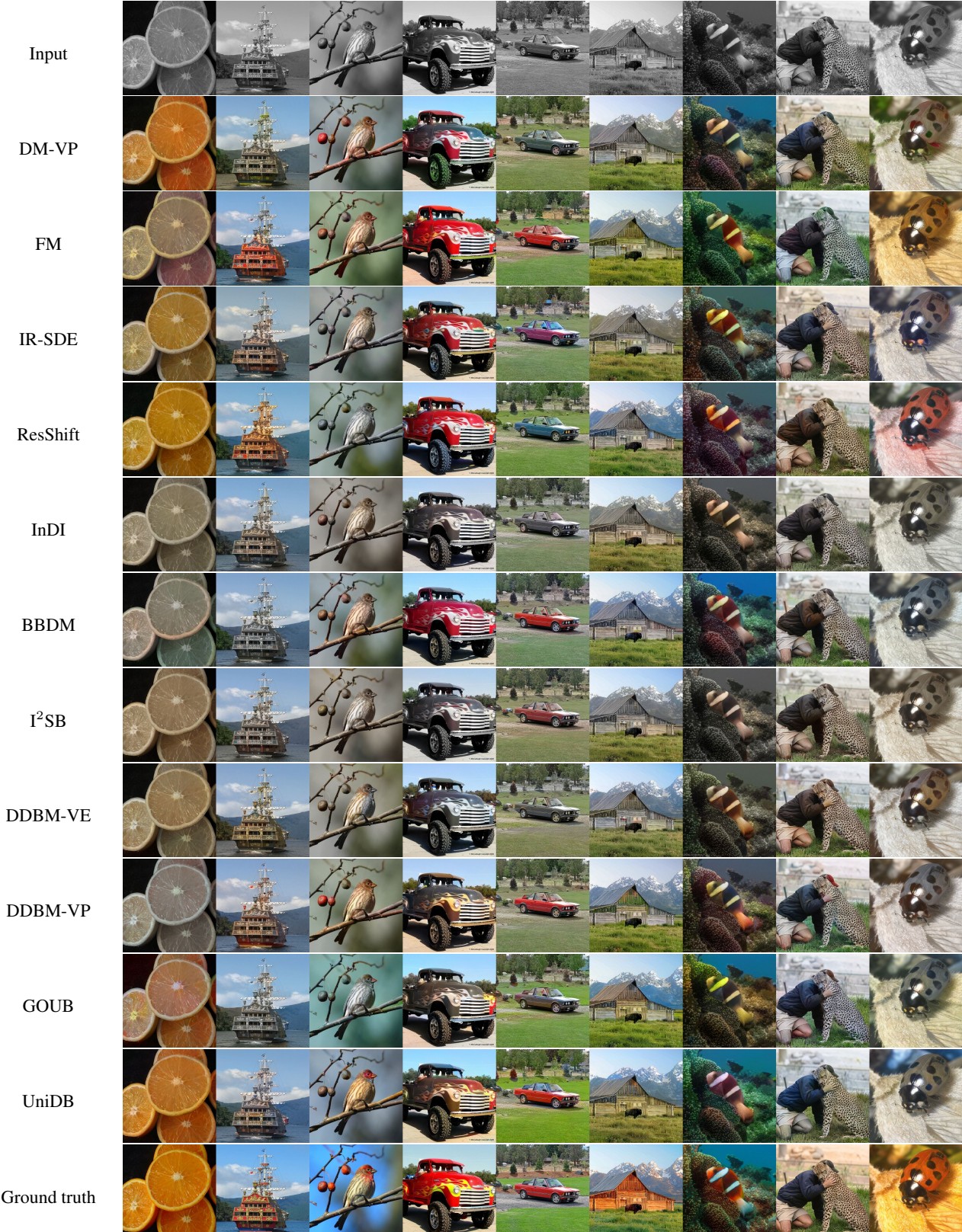

*Figure 9.* Visual comparison of image colorization on ImageNet. Results were generated using ancestral sampling with 35 steps. In most cases, methods recover the correct canonical colors (e.g., green grass, blue sky, orange fruit). However there are some exceptions, such as DDPM producing green tires and most methods struggling with colors of the ladybug. Flow Matching and ResShift produces outputs with higher saturation.

