# OpenReview forum: "Unifying Deep Stochastic Processes for Image Enhancement"
_ICML.cc/2026/Conference — ICML 2026 regular_

### Official Review · Reviewer_6FbR · 2026-03-03

**Soundness:** 3
**Presentation:** 3
**Significance:** 3
**Originality:** 3
**Overall Recommendation:** 4
**Confidence:** 3

**Summary:**

This paper presents a theoretical unification and an empirical benchmarking framework for continuous-time deep stochastic processes in image enhancement. The authors classify existing generative restoration methods into three distinct paradigms: unconditional diffusion models, Ornstein–Uhlenbeck (OU) processes, and diffusion bridges. Mathematically, the manuscript demonstrates that these diverse methods can be subsumed under a generalized Stochastic Differential Equation (SDE) framework, wherein they differ primarily in their predefined drift and diffusion terms, terminal distributions, and boundary conditions, treating samplers as orthogonal choices. To validate this unification, the authors introduce ItoVision, a modular PyTorch library designed to enforce a strictly controlled empirical evaluation. By evaluating these unified SDEs under identical network architectures and homogenized training protocols, the authors conclude that no single SDE formulation consistently dominates across image enhancement tasks. Instead, the study aims to disentangle how specific mathematical design choices empirically influence restoration performance.

**Compliance With Llm Reviewing Policy:**

Affirmed.

**Final Justification:**

After author's rebuttal, I raise my rating.

**Key Questions For Authors:**

Q1. The authors should clarify how the strong performance of Mean-ODE and Euler ODE shown in Table 14 can be reconciled with the claim in the main text that deterministic samplers lead to severe oversmoothing. Mathematical or empirical justification is required to verify whether this “oversmoothing” is merely an artifact caused by the forced L2 (MSE) loss. The results of applying Euler ODE/Mean-ODE with L1 loss for these bridge processes should also be provided.

 Q2. Comparative experiments under the original optimal configurations of each baseline method, including their dedicated loss functions, should be supplied to enable a more comprehensive and fair evaluation of their real performance.

 Q3. Empirical experiments and theoretical analysis related to MeanFlow should be incorporated into the unified framework to validate its applicability to the latest 2025 continuous-time dynamics research.

**Limitations:**

No. Recommend to include a discussion of potential failure cases and provide corresponding visual analysis to better define the boundaries of the proposed framework.

**Strengths And Weaknesses:**

Strengths:

Originality & Significance: The conceptual motivation to systematize the fragmented landscape of deep stochastic processes for image enhancement is highly relevant. The introduction of the "ItoVision" codebase provides a valuable engineering contribution, offering a standardized platform that could benefit future empirical benchmarking.

Detailed mathematical analysis: The mathematical derivations that map various conditional models (Diffusion Models, OU processes, Diffusion Bridges) onto a common SDE formulation are clearly articulated.

Phenomenological: The empirical identification of the "temperature trap"—demonstrating how low temperature ($\tau$) induces trajectory collinearity and restricts generative diversity—is a sound and useful phenomenological observation.

Weakness：

1. Limitations of the Unified Loss and Table 14 Discrepancy
Enforcing a universal L2 (MSE) loss to establish a "fair" baseline may inadvertently alter the optimization landscape of complex SDE models, masking the true capabilities of those requiring topology-aware losses like L1. This potential artifact highlights an interesting discrepancy: while the main text argues that deterministic ODE samplers inherently cause severe oversmoothing in diffusion bridges, Table 14 actually shows them achieving highly competitive super-resolution performance. I kindly encourage the authors to discuss whether this observed "oversmoothing" is merely a byproduct of the L2 loss biasing predictions toward a blurry conditional mean, and whether evaluating these samplers with an L1 loss might resolve the issue.

2. Dataset Split Issue.
In the low-light enhancement experiments (LOL dataset), the paper adopts a 495 (training) / 5 (testing) split, which deviates from the widely recognized 485/15 community standard. Evaluating the generalization capabilities of complex generative models on merely 5 test images carries limited statistical significance and may inadvertently introduce variance or accidental bias. It is recommended that the authors adhere to the standard split protocol and consider evaluating the models on more challenging, extreme low-light datasets to better validate their robustness.

3. Lack of Reproducibility and Variance Analysis
The training of stochastic generative models inherently exhibits variance. To robustly support the conclusions drawn from the "unified baseline," relying on single-run performance metrics might be susceptible to the randomness of seed initialization.

4. Discussion on Recent Frontier Advancements
While the theoretical framework nicely synthesizes existing SDE paradigms, the discussion of recent continuous-time generative breakthroughs could be expanded. For instance, "Mean Flows for One-step Generative Modeling (NeurIPS 2025)" introduces an interval-averaged velocity paradigm, and Diffusion Models are "Secretly Exchangeable: Parallelizing DDPMs via Auto Speculation (ICML 2025)" provides a fresh theoretical perspective on sampling and oversmoothing.

---

> ### Author Rebuttal · Authors · 2026-03-31
>
> We thank the reviewer for the thoughtful feedback and for recognizing the originality of our unified framework, the clarity of the mathematical formulation, the practical value of ItoVision, and the insights of our empirical findings.
>
> Below we address the concerns in detail.
>
> ## Oversmoothing
> In Table 14, diffusion bridges combined with deterministic samplers yield relatively high LPIPS values (e.g., > 0.3), despite achieving higher PSNR and SSIM. This discrepancy arises because PSNR/SSIM favor pixel-wise fidelity and tend to reward over-smoothed solutions, whereas LPIPS better captures perceptual quality and high-frequency detail preservation. We provide qualitative examples (anonymous link, https://anonymous.4open.science/api/repo/rebuttal_icml-BC4B/file/rebuttal.pdf?v=4a13aaf1) to illustrate this effect.
>
> This behavior is not only due to the L2 objective. While L1 may partially reduce smoothing, we observe that replacing deterministic samplers with stochastic ones consistently restores high-frequency details for bridge methods. This indicates that oversmoothing is primarily driven by deterministic sampling dynamics rather than the choice of loss function.
>
> ## Original configurations
> Importantly, we also report results using each method’s original configuration (including native loss functions, parameterizations, and samplers), as shown in Table 12. In this setting, each method is evaluated under its optimal design choices rather than the unified MSE objective. We observe consistent trends between the unified and original-configuration evaluations, particularly in the relative performance of diffusion, OU, and bridge processes.
> For more information, please refer to the answer to the Reviewer 8JrZ.
>
> We will clarify this more explicitly in the main text.
>
> ## Dataset Split Issue.
> We thank the reviewer for the careful analysis of our training protocol. As stated in Appendix D, “We used the original train and test splits, and we moved 5 pairs from the training set to create a validation split, as one was not provided.”. This means that the final evaluation was performed exclusively on the original “eval15” test set. The additional 5 samples taken from the training set were used only for validation during training (e.g., to select the best models with respect to the LPIPS metric). This setup avoids any data leakage while maintaining consistency with widely established evaluation practices.
>
> ## Variance Analysis.
> In our study, we report single-run results primarily due to the substantial computational cost of training a large number of models under controlled settings.
> However, we emphasize that our conclusions are based on consistent trends observed across multiple tasks, datasets, and model families, rather than isolated numerical differences. In particular, the relative behavior of diffusion, OU, and bridge processes remains stable across these settings, suggesting that our findings are robust to typical sources of stochastic variation.
> Additionally, in Table 15 we aggregate results at the level of process families, effectively averaging over multiple methods within each class. This further reduces sensitivity to individual model fluctuations and allows us to draw more reliable conclusions about the behavior of entire process families rather than specific implementations.
>
> ## Discussion on Recent Works.
> ### MeanFlow
> This study aims to provide a unified view of generative processes and evaluate their impact on result quality. Within our framework, MeanFlow would have the same definition as Flow Matching (FM), since both rely on the same underlying forward process. The main difference lies in the training objective, where MeanFlow introduces a modified loss that better captures the process dynamics. This connection is also noted in the MeanFlow paper (Section 5.1): “Our method can be viewed as Flow Matching with a modified target...”. We will cite this work in section 2.2 and compare it with FM, similarly to how we did for Rectified Flow.
>
> ### Secretly Exchangeable: Parallelizing DDPMs via Auto Speculation.
> The contribution of this work does not lie in defining a new stochastic process, but in proposing an alternative sampling procedure and discretization of the transport dynamics. It modifies how trajectories between distributions are approximated, while keeping the underlying generative process unchanged.
>
> In our framework, such methods correspond to different samplers or parameterizations of the same process, and are therefore orthogonal to the methods we study, which focus on differences between process families (diffusion, OU, and bridge processes) defined at the level of the forward SDE and transition kernels.
>
> We also plan to cite and incorporate it into our library as a sampler once it is published.
>
> We hope that our responses have addressed the reviewer’s concerns and clarified the contributions and significance of our work. If so, we would kindly ask the reviewer to reconsider their score.

---

> > ### Author Rebuttal · Reviewer_6FbR · 2026-04-03
> >
> > I have carefully read the authors' rebuttal and appreciate the clarifications regarding the dataset split and the inclusion of original configurations in Table 12. However, after further consideration, my core concerns regarding the methodological validity of the empirical benchmarking and the framework's true innovativeness remain unresolved.
> >
> >
> > 1. The "Unified Baseline" fundamentally dismantles modern theoretical constructions.
> > The authors argue that standardizing the models under a unified MSE objective and identical network architectures provides a "fair" comparison. However, in contemporary continuous-time generative research, a model's efficacy is rarely derived from the SDE formulation alone. It is heavily reliant on the tight, theoretically justified coupling between the forward process, specialized topology-aware losses (e.g., frequency-domain or structural losses), and tailored samplers. By stripping these methods of their dedicated components and forcing them into a primitive, earliest-stage configuration, the unified baseline does not establish fairness; rather, it actively breaks the theoretical constructions that make these models viable in practice. This limits the practical significance of the benchmark.
> >
> >
> > 2. Lack of empirical validation for recent advancements.
> > While the authors offer theoretical explanations for how 2025-2026 methods like "MeanFlow" or "Secretly Exchangeable DDPMs" could fit into their SDE taxonomy (treating them merely as modified targets or orthogonal samplers), theoretical taxonomy is insufficient for a paper whose main contribution hinges on an empirical benchmarking library. Without concrete empirical experiments integrating and validating these recent continuous-time dynamics breakthroughs, the claim that this framework robustly encapsulates the current frontier of image enhancement remains unconvincing.
> >
> >
> > Given these fundamental methodological limitations, the paper's contribution leans more towards a retrospective taxonomy rather than a forward-looking or practically innovative framework.
> >
> > After author's rebuttal, I will raise my rating.

---

> > > ### Author Response · Authors · 2026-04-07
> > >
> > > We thank the reviewer for detailed feedback.
> > >
> > > First, we kindly disagree that the primary contribution of this work is empirical benchmarking. Rather, our main contribution is the unification of diverse methods under a common stochastic process framework, with benchmarking serving as a secondary component to study the effect of these processes.
> > > In this context, our focus is specifically on the stochastic processes underlying these methods, and we aim to isolate their impact on performance. To enable such a controlled and meaningful comparison, we keep all other components, such as architectures, losses, and training setups, fixed across methods.
> > >
> > > At the same time, we agree that evaluating complete methods (i.e., including their dedicated losses, samplers, and other components) is important. For this reason, we additionally trained and reported results for the original configurations of the methods, and we included corresponding analysis and discussion in the paper.
> > >
> > > We do not dispute that incorporating specific components, such as L1 or other specialized losses, can improve performance for certain methods. However, this raises a broader question: such modifications could potentially benefit other models as well, making it difficult to attribute improvements to a specific design choice. This applies not only to losses but to many aspects of these methods. Therefore, in our controlled comparison, we intentionally limit variation to the stochastic processes themselves, while separately reporting results for original configurations.
> > >
> > > Finally, regarding recent works such as Mean Flow or Secretly Exchangeable DDPMs, we do not claim that they are merely samplers or losses. On the contrary, we agree that all components of these methods are important. However, within the scope of this work, we focus specifically on analyzing how the underlying stochastic processes influence model behavior and performance. Since these methods do not correspond to separate SDE definitions in the same sense as those included in our unified formulation, we cannot include them in the theoretical comparison in Table 1, which focuses on SDE definitions and transition kernels unified under our framework, nor in Table 2, which compares different SDE formulations across tasks. Nevertheless, we can include MeanFlow in Table 12 and Secretly Exchangeable DDPMs in Tables 13 and 14. For Mean Flow, we have so far successfully trained the original implementation on the super-resolution task using the FFHQ dataset, and we provide results in the table below. As we can see Mean Flow gives competitve results to Flow Matching (FM) but for higher Network Forward Evaluations (NFE) it does not scale as good as original FM. We plan to expand the experiments on these methods and update the paper with appropriate conclusions once all experiments are completed.
> > >
> > > | NFE | Method | PSNR | SSIM | LPIPS | NIQE | FID |
> > > |-----|--------|------|------|-------|------|-----|
> > > | **1**   | Flow Matching | **29.21** | **0.786** | 0.281 | 7.50 | **18.47** |
> > > |         | Mean Flow     | 28.32     | 0.766     | **0.279** | **7.13** | 18.96     |
> > > |-----|-----------------------|--------|--------|---------|------|-------|
> > > | **5**   | Flow Matching | **28.47** | **0.769** | **0.233** | 6.09 | **17.35** |
> > > |         | Mean Flow     | 26.68     | 0.732     | 0.247     | **5.40** | 18.87     |
> > > |-----|-----------------------|--------|--------|---------|------|-------|
> > > | **35**  | Flow Matching | **27.01** | **0.716** | **0.169** | **3.96** | **17.40** |
> > > |         | Mean Flow     | 24.53     | 0.663     | 0.237     | 4.59     | 20.03     |
> > > |-----|-----------------------|--------|--------|---------|------|-------|
> > > | **100** | Flow Matching | **26.70** | **0.697** | **0.168** | **3.63** | **17.57** |
> > > |         | Mean Flow     | 24.19     | 0.643     | 0.243     | 4.73     | 20.07     |

---

### Official Review · Reviewer_6ppk · 2026-03-09

**Soundness:** 3
**Presentation:** 3
**Significance:** 3
**Originality:** 2
**Overall Recommendation:** 5
**Confidence:** 3

**Summary:**

This paper provides an unified perspective on various stochastic image enhancement processes. Specifically, this work focuses on three classes of stochastic processes: standard diffusion, OU process, and diffusion bridges. It is shown that these approaches can be explicitly derived from a common SDE. Then this paper conducts an empirical study across different models and multiple image enhancement tasks, to identify and disentangle specific design choices of the stochastic process. In addition, this work contributes a modular PyTorch library, called ItoVision, for fast implementation and comparison of different stochastic image enhancement algorithms.

**Compliance With Llm Reviewing Policy:**

Affirmed.

**Final Justification:**

The authors well addressed my questions and concerns. Thus I increased my score to accept.

**Key Questions For Authors:**

Have the authors tried to compare different stochastic methods on image deblurring (such as on the GoPro dataset), which in my experience is more sensitive to the noise temperature.

**Limitations:**

As I mentioned in the **Weaknesses**, this paper itself doesn't propose a new algorithm which might limits its novelty.

**Strengths And Weaknesses:**

**Strengths**
1. The motivation is strong to me. I agree that most current stochastic image enhancement methods are relevant but haven't been unified under the same framework, thus it's hard to construct a fair comparison.
2. The main contribution of this paper is not a new method. Instead, it unifies different stochastic processes into a common SDE framework. This work further decouples these stochastic methods from their schedulers and samplers.
3. With the unified framework, this paper conducts experiments across multiple image restoration tasks for a fair comparison.
4. The resulting framework is both practically and theoretically important to image restoration research area. The release of a modular PyTorch library further strengthens reproducibility.

**Weaknesses**
1. My main concern is that this paper itself doesn't propose a new algorithm. The novelty is more conceptual and experimental than algorithmic.
2. In Eq. (3), it would be better to add a proof or explanation that how $\lambda$ is added to the diffusion coefficient.
3. For stochastic method, it would be interesting to see the authors further discuss stochastic interpolants[1] and forward-only diffusion (i.e., OU in both drift and diffusion terms[2]).

[1] Stochastic interpolants: A unifying framework for flows and diffusions. JMLR 2025.
[2] Forward-only diffusion probabilistic models. arxiv 2025.

---

> ### Author Rebuttal · Authors · 2026-03-31
>
> We thank the reviewer for the constructive feedback and for recognizing the value of our unified SDE framework, fair comparison, and ItoVision library. Below we address the concerns in detail.
>
> ## Novelty beyond algorithmic contribution.
> We would like to emphasize both its theoretical and empirical novelty. On the theoretical side, we provide a unified SDE formulation that places previously disparate methods, diffusion, OU processes, and diffusion bridges, within a single mathematical framework, explicitly characterizing their differences in terms of drift, diffusion, and boundary conditions. This perspective clarifies relationships that were previously implicit and fragmented across the literature. On the empirical side, this unification enables the first controlled comparison across methods under identical conditions, leading to new insights (e.g., the limited advantage of conditional trajectories and the role of temperature). We will clarify this positioning more explicitly in the paper.
>
> ## How is $\lambda$ added to the diffusion coefficient.
> While this is a well-established result and not central to our main contributions, it remains important for the formulation of the reverse process. We note that the proof is available in prior work [1], which we already cite next to Eq. (3). For additional clarity, we will include a remark after Eq. (3) stating “for a complete proof, see [1], Appendix H.” to direct the reader to the exact location of the derivation.
>
> Currently, the Appendix contains only proofs of our original contributions. However, if the reviewer believes it would be beneficial, we are happy to include this proof in our Appendix.
>
> [1] Zhang, Qinsheng, and Yongxin Chen. "Diffusion normalizing flow." Advances in neural information processing systems 34 (2021): 16280-16291.
>
> ## Relation to stochastic interpolants and forward-only diffusion.
>
> ### Stochastic Interpolants
> We thank the reviewer for pointing out this important work on the unification of Flow Matching and diffusion. We will include it in both the Related Work and the Introduction, as it shares conceptual similarities with our study.
>
> At the same time, we would like to highlight several key differences. First, the referenced work focuses specifically on unifying Flow Matching and diffusion, whereas our study considers a broader class of conditional processes designed specifically for Image Enhancement, such as Ornstein–Uhlenbeck processes and diffusion bridges. Second, although the prior work formulates Flow Matching in terms of SDEs, it only considers a linear rate of change. In contrast, our formulation generalizes this to arbitrary schedulers, which still define interpolation between prior and target distributions but do not require the rate of change to be linear in time.
>
> ### Forward-only diffusion
> This paper proposes a forward process that transforms the prior distribution into the target distribution. In our framework, such a process would correspond to the reverse process, and it would require identifying an SDE whose reversal (via Eq. (3)) recovers their original formulation.
>
> We will include this work in our paper. However, to ensure the correctness of our claims, we require additional time to carefully analyze, unify, and properly position this approach within our taxonomy. We will make our best effort to provide the formulation as soon as possible.
>
> ## Deblurring experiments.
> We agree that image deblurring is a relevant and challenging task. To test the hypothesis that temperature plays a particularly important role in this setting, we trained four models: InDI with low (0.06) and high (2.0) temperatures, as well as DDPM and I2SB, representing unconditional and bridge diffusion methods, respectively.
>
> The results (shared via the anonymous link, https://anonymous.4open.science/api/repo/rebuttal_icml-BC4B/file/rebuttal.pdf?v=4a13aaf1) show that InDI trained with a low temperature achieves the worst performance and exhibits significant training instability. This observation is consistent with the reviewer’s hypothesis and aligns with our findings regarding the impact of temperature. Below we present the results for NFE=35. In the anonymous repository we also provide results for different NFEs.
>
> | Metric        | DDPM | I²SB | InDI | InDI high τ |
> |--------------|------|------|------|-------------|
> | PSNR ↑       | **29.08** | 28.99 | 27.32 | 28.98 |
> | SSIM ↑       | 0.843 | **0.844** | 0.793 | 0.841 |
> | LPIPS ↓      | 0.111 | **0.103** | 0.119 | 0.106 |
> | NIQE ↓       | 4.64 | 4.74 | **4.17** | 4.56 |
> | CLIP-IQA ↑   | **0.362** | 0.354 | 0.281 | 0.343 |
> | MusiQ ↑      | 43.74 | **45.56** | 42.58 | 44.28 |
>
> We will include the full experiment, along with all methods, in the final version of the paper.
>
> We hope that our responses have addressed the reviewer’s concerns and clarified the contributions and significance of our work. If so, we would kindly ask the reviewer to reconsider their score.

---

> > ### Author Rebuttal · Reviewer_6ppk · 2026-04-02
> >
> > I thank the authors for their efforts in the rebuttal. I am quite positive to this paper and the framework itself. However, I still have a question/concern regarding the algorithmic contribution. While all results from the paper are built on existing algorithms (via the unified SDE framework), it would be nicer if this work can present a new algorithm that lies in the same SDE framework but outperforms (or is more effective than) other existing methods.

---

> > > ### Author Response · Authors · 2026-04-06
> > >
> > > We thank the reviewer for their positive assessment of our paper and framework and appreciate the thoughtful feedback.
> > >
> > > ## On the Importance of Systematic Understanding in Generative Modeling
> > > To provide additional context, this work aligns with a line of research that focuses on uncovering the fundamental properties of methods and understanding their impact on performance, as a complementary direction to developing new state-of-the-art approaches. We believe this perspective can significantly contribute to the field’s progress.
> > >
> > > Several influential works follow this paradigm. For example, "Stochastic Interpolants", cited by the reviewer, and "Are GANs Created Equal? A Large-Scale Study" [1] are impactful not because they introduce new methods, but due to their theoretical unification and systematic empirical comparison, respectively.
> > >
> > > In particular, "Stochastic Interpolants" have shaped subsequent work by enabling flexible design of generative processes (e.g., controllable noise, connections to Schrödinger bridges) and by clarifying how different modeling choices correspond to trajectories within a common SDE framework.
> > >
> > > Similarly, "Are GANs Created Equal?" is highly influential as it established the importance of systematic and fair empirical evaluation in generative modeling, demonstrating that many GAN variants achieve comparable performance when properly tuned.
> > >
> > > ## Contributions of the Unified SDE Framework
> > > Our work is guided by the same philosophy, integrating theoretical unification with systematic empirical evaluation. While this work does not introduce new approach, we argue there are several benefits of this study including:
> > >
> > > (i) it shows the true differences between method definitions - through Propositions 2.1-2.3 we obtain method definitions that can be directly compared in Table 1,
> > > (ii) Common mathematical framework allows fair comparison using the same solvers and discretization techniques, and explains why some methods perform better than others (e.g., issues with low temperature discussed in the paper),
> > > (iii) it provides unified source code, which can support further improvements in the field,
> > > (iv) without such unification, prior works rarely compare methods directly. For example, ResShift acknowledges methods like InDI but treats them as separate due to different formulations, and in GOUB, the methods BBDM and I2SB, although cited as instances of Brownian and Schrödinger Bridges, were never directly compared to other diffusion bridge methods, as they were considered part of a different framework. Our contribution shows that these models can be compared fairly and, in some cases, are even equivalent (e.g. BBDM, DDBM-VE, I2SB).
> > >
> > > ## New processes given the Unified Framework
> > > We also would like to emphasize that, given the unified perspective, it becomes straightforward to construct new processes with modified properties, e.g. by defining continuous-time transition kernels and differentiating them to obtain SDE coefficients, or by applying Doob’s h-transform to form bridge processes.
> > >
> > > However, our empirical results suggest that developing increasingly complex process variants may not be the most effective research direction, as such methods rarely outperform strong baselines (e.g. diffusion models) when training is standardized. Instead, our goal is to explain why these models can underperform, thereby guiding future work toward the factors that have the most significant impact on performance.
> > >
> > > ## Forward-diffusion additional theory
> > > Additionally, we provide an update on Forward-only Diffusion. We classify it as a diffusion bridge since the marginal variance vanishes at both endpoints. However, because the diffusion coefficient depends on the state $x_t$, the transition kernels are not Gaussian (unlike in prior work), but log-Gaussian shifted by $\mu$; specifically, the residuals $e=\mu-x_t$ follow a Geometric Brownian motion. The process remains reversible in the Anderson sense, yielding the forward SDE
> > > $$dx_t=[e(\theta_t+\sigma_t^2(2-e\nabla_{x_t}\log p_t(x_t | y, \mu)))]dt-\sigma_t e\;d\bar{w}_t.$$
> > >
> > > Reversing this recovers Eq. (6) of the Forward-only Diffusion paper. A key property is that trajectories never cross the initial condition $\mu$ (i.e. $\lnot\exists_{s,t}:(x_s<\mu<x_t)$ for each pixel). This constraint may be harmful: given $y$ and $x_t$, the model learns that $x_0$ ($\mu$) cannot lie between them, forcing extrapolation along $x_t − y$, similar to low-temperature models (e.g., InDI). On the other hand, this dynamic may encourage conservative updates, reducing overshooting of $x_0$.
> > >
> > > We are currently conducting empirical validation and will include results in the camera-ready version. Note that the original implementation does not condition the backbone on y. We will train both variants: the original and a conditioned version, as unconditional backbones perform significantly worse (I2SB, Table 12).
> > >
> > > [1] Lucic et al., Are GANs created equal? A Large-Scale Study, NeurIPS 2018.

---

### Official Review · Reviewer_8JrZ · 2026-03-12

**Soundness:** 4
**Presentation:** 4
**Significance:** 4
**Originality:** 4
**Overall Recommendation:** 5
**Confidence:** 4

**Summary:**

This paper presents a unified mathematical perspective on stochastic image enhancement by classifying recent methods into three continuous-time families: unconditional diffusion models, Ornstein-Uhlenbeck (OU) processes, and diffusion bridges. The authors demonstrate that these seemingly disparate approaches can be formulated under a common stochastic differential equation (SDE) framework. The framework highlights that these methods differ primarily in drift/diffusion terms and boundary conditions, treating schedulers, samplers, and discretization strategies as orthogonal design choices.

**Compliance With Llm Reviewing Policy:**

Affirmed.

**Final Justification:**

After reviewing the authors’ rebuttal, I am maintaining my original positive evaluation and scores.
The authors provided clear, detailed, and convincing responses to the majority of my concerns.

**Key Questions For Authors:**

1. In your unified controlled study, you fixed the objective to the MSE loss for all methods to isolate the effect of the underlying processes. However, some methods natively utilize different loss functions in their original implementations. Could forcing MSE artificially disadvantage methods that theoretically or empirically require different objectives to model their specific transition kernels optimally?
2. The finding that standard, unconditional diffusion often matches or outperforms conditional processes is striking. Does the unified SDE framework suggest that conditional processes are fundamentally bounded by the same capacity as standard diffusion, or do you hypothesize there are specific extreme degradation spaces where explicit trajectory conditioning would demonstrate a clear theoretical and empirical edge?
3. Could you elaborate on why DiT architectures exhibit a notable drop in performance as NFE increases (as detailed in Appendix G), contrary to the stable scaling behavior observed with UNet? Is this an artifact of the specific DiT scale used, or a broader characteristic of transformer backbones operating within these unified stochastic processes?
4. Standard metrics like LPIPS, NIQE, and FID can be limited in capturing fine-grained quality and semantic consistency for highly ill-posed tasks. Could the authors incorporate modern no-reference metrics (e.g., CLIP-IQA or MUSIQ) to better substantiate the aesthetic and semantic fidelity of the results?

**Limitations:**

Yes

**Strengths And Weaknesses:**

Strengths:
1. The conceptual unification of discrete Markov chains, flow-based models, and Schrödinger/Brownian bridges into a cohesive continuous-time SDE framework is mathematically rigorous.
2. The release of the ItoVision library provides a practical asset for the community's reproducibility and future methodological development.

Weaknesses:
1. While the mathematical derivations in the appendix are detailed, the main text could benefit from better structural clarity when mapping discrete schedulers to continuous SDEs (e.g., Section 2.3 for ResShift). Integrating  alongside the mathematical transitions could significantly aid reader intuition.
2. The main empirical conclusions are drawn almost exclusively from UNet backbones. While the authors provide Diffusion Transformer (DiT) experiments in Appendix G, they note a degradation in DiT performance as Network Forward Evaluations (NFE) increase, which sharply contrasts with UNet behavior. Expanding upon this architectural discrepancy in the main text would strengthen the study's comprehensiveness.

---

> ### Author Rebuttal · Authors · 2026-03-31
>
> We thank the reviewer for the positive assessment of our work and for the insightful suggestions.
>
> ## MSE objective and fairness of comparison.
>
> Importantly, we also report results using each method’s original configuration (including their native loss functions, parameterizations, and samplers), as presented in Table 12. In this setting, each method is evaluated under its original design choices rather than the unified MSE objective. We observe consistent trends between the unified setup (Table 2) and the original-configuration evaluations (Table 12, NFE=35), particularly in the relative behavior of diffusion, OU, and bridge processes.
>
> While individual methods may benefit from their native configurations and exhibit shifts in specific metrics (e.g., PSNR/SSIM vs LPIPS/FID trade-offs), the overall conclusions remain unchanged.
>
> As examples, diffusion-based models (especially Flow Matching) achieve strong perceptual performance (low LPIPS and FID) in both Table 2 and Table 12. Similarly, diffusion bridge methods (e.g., GOUB, UniDB) tend to achieve strong distortion metrics (PSNR/SSIM), while trading off perceptual quality (higher LPIPS/FID), consistent with the unified setting and reflecting a bias toward pixel-wise fidelity.
>
> Notably, OU-based methods such as IR-SDE consistently exhibit worse perceptual quality (e.g., high LPIPS and FID), despite using their native configurations.
>
> To further isolate component effects, we include additional experiments (see Table 1 in the anonymous repository, https://anonymous.4open.science/api/repo/rebuttal_icml-BC4B/file/rebuttal.pdf?v=4a13aaf1) comparing L1 vs L2 objectives and ancestral vs Euler ODE samplers for InDI. While L1 improves perceptual quality, the gains are smaller compared to the effect of temperature.
>
> Increasing the temperature parameter $\tau$ yields substantially larger improvements across both perceptual and distortion metrics, consistently outperforming both L1 and standard L2. This highlights that process-level factors, such as stochasticity and trajectory geometry, have a stronger impact than the choice of loss.
>
> ## Conditional processes advantages
>
> In this work, we indeed demonstrate that, under controlled experimental settings, conditional processes do not consistently outperform standard diffusion models. To better understand this phenomenon, we note that even in unconditional processes such as diffusion, where the trajectory itself does not depend on y, the backbone model still receives y as an additional input.
>
> We hypothesize that this conditioning provides a very strong signal, which makes the specific choice of trajectory less critical, even in the presence of severe degradations. This is supported by the I2SB results reported in Table 12, where we reproduce the original setting in which I2SB does not condition on y. The observed performance suggests that the absence of such conditioning plays a more significant role than the trajectory design itself, indicating that the different methods have comparable capacity within this setup.
>
> At the same time, we believe there is still meaningful potential for conditional processes in unsupervised settings, where paired data are not available and training a conditional backbone is therefore not feasible, as in the case of Schrödinger Bridge methods.
>
> ## DiT problems
> We believe that DiT models are powerful backbones for diffusion-based architectures, but they require a large number of meaningful tokens to operate effectively. This often leads to tokenizations at the level of one or a few pixels per token. When DiT is applied directly in pixel space rather than in the latent space of a VAE, each token carries very limited semantic information.
>
> Our hypothesis is that DiT models need to operate in a latent space and be trained on large-scale datasets in order to fully capture the complex relationships between tokens. This may explain why models such as Stable Diffusion 3.5 or Flux achieve strong results, while smaller DiT-based models can struggle in comparison to UNet-based architectures.
>
> ## Additional metrics
> We included CLIP-IQA and MusiQ metrics in our comparisons. The updated results from each task is sent via anonymous link. While these metrics correlate with measures such as LPIPS and FID, they strengthen the evaluation by offering additional validation of the results.
>
> An important remark regarding CLIP-IQA is that images must be resized to 224 pixels for proper evaluation. For tasks such as super-resolution from 64 to 512, this resizing can remove fine-grained details produced by the model. This can cause lower variance of this metric than MUSIQ, and it should be interpreted with caution.
>
> ## Mapping discrete schedulers to continuous SDEs.
> We agree that some derivations may be unclear in places due to page length constraints. Since the camera-ready version allows for an additional page, we will enrich the derivation in the main text with more intuitions.

---

### Decision · Program_Chairs · 2026-04-30

**Decision:**

Accept (regular)

**Comment:**

This paper presents a unified stochastic differential equation (SDE) perspective on stochastic image enhancement, showing that several families of methods—standard diffusion, Ornstein–Uhlenbeck processes, and diffusion bridges—can be formulated within a common continuous-time framework. The paper further introduces a modular library, ItoVision, and uses the unified formulation to conduct controlled empirical comparisons across methods and tasks.

After reading the reviews, rebuttal, and follow-up discussion, my overall assessment is positive, and I recommend acceptance.